# Unlocking the Power of SAM 2 for Few-Shot Segmentation

**Qianxiong Xu** [1]   **Lanyun Zhu** [2]   **Xuanyi Liu** [3]   **Guosheng Lin** [1]   **Cheng Long** [1]   **Ziyue Li** [4]   **Rui Zhao** [5]

## Abstract

Few-Shot Segmentation (FSS) aims to learn class-agnostic segmentation on few classes to segment arbitrary classes, but at the risk of overfitting. To address this, some methods use the well-learned knowledge of foundation models (e.g., SAM) to simplify the learning process. Recently, SAM 2 has extended SAM by supporting video segmentation, whose class-agnostic matching ability is useful to FSS. A simple idea is to encode support foreground (FG) features as memory, with which query FG features are matched and fused. Unfortunately, the FG objects in different frames of SAM 2's video data are always the same identity, while those in FSS are different identities, i.e., the matching step is incompatible. Therefore, we design Pseudo Prompt Generator to encode pseudo query memory, matching with query features in a compatible way. However, the memories can never be as accurate as the real ones, i.e., they are likely to contain incomplete query FG, and some unexpected query background (BG) features, leading to wrong segmentation. Hence, we further design Iterative Memory Refinement to fuse more query FG features into the memory, and devise a Support-Calibrated Memory Attention to suppress the unexpected query BG features in memory. Extensive experiments have been conducted on PASCAL-$5^i$ and COCO-$20^i$ to validate the effectiveness of our design, e.g., the 1-shot mIoU can be 4.2% better than the best baseline.

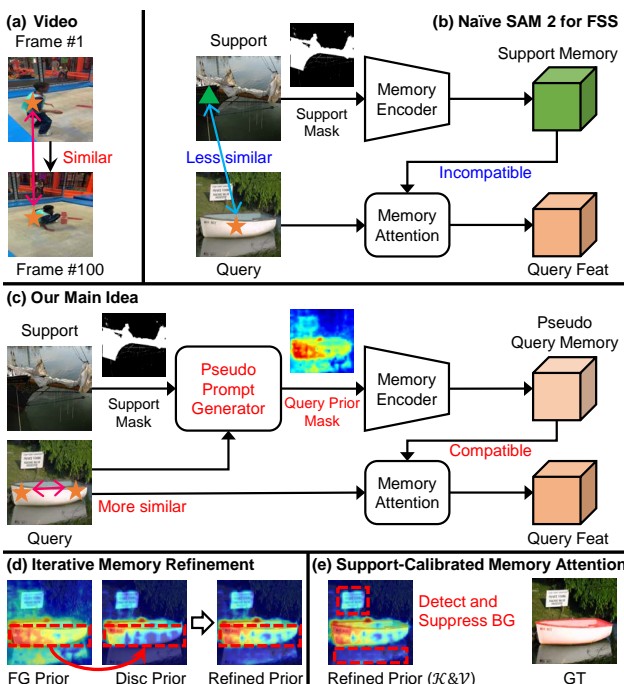

*Figure 1.* Illustrations of (a) video data, (b) simple use of SAM 2, (c) our main idea, (d) prior masks and our Iterative Memory Refinement, and (e) our Support-Calibrated Memory Attention. In (a), SAM 2's learned knowledge is same-objects matching. In (b), the objects in FSS are *different*, posing challenges to use SAM 2's knowledge. In (c), we generate prior masks and encode them as pseudo query memories to enable same-objects matching. In (d) and (e), we use priors to visualize memories, and fuse more query FG, while suppress the unexpected query BG features in memory.

## 1. Introduction

Semantic segmentation is a fundamental vision task that involves assigning class labels to each pixel in an image, which is essential for detailed scene understanding, and has

[1]S-Lab, Nanyang Technological University [2]Singapore University of Technology and Design [3]Peking University [4]University of Cologne [5]SenseTime Research. Correspondence to: Guosheng Lin <gslin@ntu.edu.sg>, Cheng Long <c.long@ntu.edu.sg>.

*Proceedings of the 42$^{nd}$ International Conference on Machine Learning*, Vancouver, Canada. PMLR 267, 2025. Copyright 2025 by the author(s).

seen significant advancements (Long et al., 2015). Unfortunately, most of them depend on extensive pixel-wise annotations, requiring massive time and human efforts. Besides, they cannot generalize to novel classes that were not present during training, struggling to segment arbitrary classes.

Few-Shot Segmentation (FSS) (Shaban et al., 2017) has been introduced to address these issues, which can segment arbitrary classes by referring to a few annotated samples with the target class. The rationale lies in similarity-based feature matching, i.e., find matched objects in a query image, whose features are similar to that of annotated support samples. FSS would learn such pattern from some base classes,

and apply it to segment novel classes without re-train or fine-tune, making it a cheaper and generalizable solution.

Existing methods roughly include prototypical (Tian et al., 2020) and attention-based methods (Xu et al., 2023). Prototypical methods compress support foreground (FG) features into a few prototypes, and segment query image via feature comparison (Wang et al., 2019) or fusion (Zhang et al., 2019b). Attention methods (Zhang et al., 2021) employs cross attention to measure the feature similarity between query and support FG pixels, dynamically fuse the latter's features to the former. However, novel classes are unknown until testing, thus, these models can easily get *overfitting* on base classes, hindering the use of more parameters to learn better class-agnostic segmentation knowledge.

To tackle this issue, recent advancements (Sun et al., 2024; Zhang et al., 2024) deploy foundation models (e.g., DINOv2 (Oquab et al., 2023) and SAM (Kirillov et al., 2023)) to benefit from their fine-grained image features or the promptable image segmentation capability, making the learning of FSS easier. Recently, SAM 2 (Ravi et al., 2024) extends SAM by enabling promptable video segmentation, and has shown remarkable FG-FG matching ability among video frames, which may benefit FSS.

As shown in Figure 1(b), SAM 2 can be directly applied by encoding support features (frame #1) and its mask (prompt) as support memory, used to dynamically match and fuse query FG features (frame #2). However, the FG-FG matching of FSS is *incompatible* with that of SAM 2, as the training videos (Figure 1(a)) of SAM 2 always have underline{uniform} FG objects (e.g., human) across frames, yet those in query and support images in FSS are always *different* (e.g., ships). As an object is essentially more similar to itself rather than others, the FG-FG matching in videos is naturally easier than that in FSS data, i.e., FSS needs to re-learn these parameters to facilitate FG-FG matching and fusion between different objects. As a result, SAM 2's robust parameters will *degenerate* back to narrow ones, i.e., overfit base classes.

In this paper, we present **Few-Shot Segment Anything Model (FSSAM)** to appropriately use the matching ability of SAM 2, i.e., convert the matching between *different objects* (support&query) to that between same objects (query). The question is how to construct prompts for encoding pseudo query memories. As learning-agnostic prior masks (Tian et al., 2020) can roughly locate query FGs by measuring feature similarities between query and support FG pixels, we design **Pseudo Prompt Generator (PPG)** (Figure 1(c)) to generate and take prior masks as pseudo mask prompts. However, existing prior masks, affected by feature ambiguity (Xu et al., 2024a), will *activate many query background (BG) regions* (Figure 1(d)), i.e., the encoded memories will include many query BG features. Since the matching ability of SAM 2 is quite

strong, these BG features may lead to wrong segmentation of query BG objects. Therefore, we follow (Xu et al., 2024a) to suppress them and additionally generate discriminative (Disc) prior masks and memories. As shown in Figure 1(d) and Figure 5(b), existing FG priors usually include more complete FG but activate *more BG* regions, and Disc priors have *less complete FG* yet activate underline{less BG} regions.

As Disc memory includes incomplete query FG and few query BG features, hindering FG-FG matching, we further design **Iterative Memory Refinement (IMR)** and **Support-Calibrated Memory Attention (SCMA)** modules as follows. As displayed in Figure 1(d), we design **IMR** to complement query FG features from FG memory to Disc memory, where the support information is used to restrain the propagation of query BG features. Hence, Disc memory would have more complete query FG features, making the matching with FG objects easier. After IMR, Disc memory may include some BG features, which (1) are initially contained in Disc memory, and (2) are fused from FG memory. Therefore, we further devise a **SCMA** (Figure 1(e)) to calibrate the cross attention scores between query features and the memory, where the BGs in memory will be detected and their scores will be suppressed. Thus, query features are fused with less BG features, leading to accurate predictions.

Our contributions include: (1) We are the first to adapt SAM 2 for FSS, benefiting from its remarkable matching ability. We start with a simple way, and identify the *incompatible matching*. (2) We design **PPG** to encode pseudo query memories, enabling compatible FG-FG matching, aligned with SAM 2's well-learned parameters. (3) Pseudo query memories always include incomplete FG and unexpected BG features, so we design **IMR** to enrich query FG features, and **SCMA** to suppress the matching and fusion with query BG features. (4) Extensive experiments have been conducted to validate the effectiveness of our design. We set new state-of-the-arts on PASCAL-$5^i$ and COCO-$20^i$, e.g., the 1-shot mIoU scores are 81.0% and 62.3%, respectively.

## 2. Related Work

**Classical FSS.** Based on the way of utilizing support samples, existing methods can be categorized into prototypical (Shaban et al., 2017; Wang et al., 2019; Tian et al., 2020; Fan et al., 2022; Wang et al., 2024a), attention-based (Zhang et al., 2019a; 2021; Wang et al., 2020; Hu et al., 2019; Xu et al., 2023; Hong et al., 2022; Xiong et al., 2022; Shi et al., 2022; Liu et al., 2023a; Park et al., 2024; Iqbal et al., 2022; Peng et al., 2023; Wang et al., 2023a; Xu et al., 2024a), mamba-based (Xu et al., 2024b), and text-based methods (Yang et al., 2023; Zhu et al., 2024; Chen et al., 2024; Wang et al., 2024b). Prototypical methods usually extract a few prototypes from support features, then either measure pixel-wise feature similarity (Fan et al., 2022) or

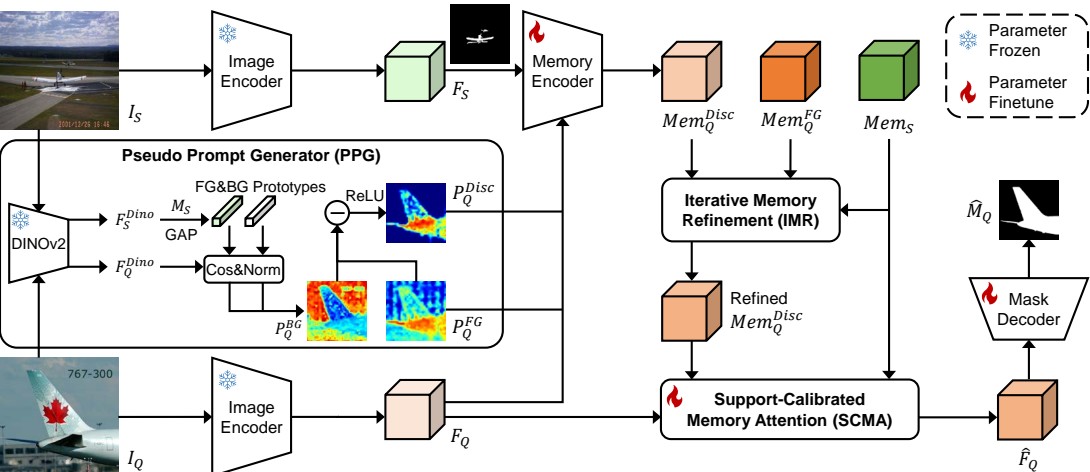

*Figure 2.* Overview of **FSSAM**, which includes: (1) **Pseudo Prompt Generator** generates a pair of FG (with more complete FG but *more wrongly activated BG* regions) and Disc priors (with *less complete FG* yet less wrongly activated BG regions) for encoding pseudo query memories $Mem_Q^{FG}$ and $Mem_Q^{Disc}$ ; (2) **Iterative Memory Refinement** aims to iteratively complement FG features from $Mem_Q^{FG}$ to $Mem_Q^{Disc}$; and (3) **Support-Calibrated Memory Attention** helps to suppress the unexpected BG features in refined $Mem_Q^{Disc}$.

fuse (Tian et al., 2020) with query features to segment FG objects. Nevertheless, the compression of support features to prototypes would inevitably result in information loss, so attention-based methods propose to take advantages of cross attention (Vaswani et al., 2017) to dynamically fuse each query pixel with uncompressed support FG features. As the computational complexity of attention is quadratic to feature sizes, HMNet (Xu et al., 2024b) design a cross attention-like mamba (Gu & Dao, 2023) to take the role of attention, but keep the complexity linear. These methods uniformly learn segmentation knowledge from a few base classes, and can easily get overfitting on them, failing to well segment arbitrary novel classes.

**Foundation-based FSS.** Some methods (Liu et al., 2023b; Sun et al., 2024; Chang et al., 2024; Zhang et al., 2024) have proposed to incorporate foundation models (e.g., DI-NOv2 (Oquab et al., 2023), SAM (Kirillov et al., 2023)) into FSS, using their well learned knowledge to simplify the learning of FSS. Particularly, SAM has shown excellent class-agnostic ability of promptable image segmentation, e.g., if any prompts like points or bounding boxes are provided, the relevant objects can be well segmented. Some efforts have been made to leverage such ability, e.g., VRP-SAM (Sun et al., 2024) designs a Visual Reference Prompt Encoder to generate prompt embeddings for SAM. Besides, GF-SAM (Zhang et al., 2024) leverages graph analysis to extract point prompts for SAM. Recently, SAM 2 (Ravi et al., 2024) has extended SAM by enabling promptable video segmentation, whose FG-FG matching ability is excellent, so we propose to leverage such ability in this paper.

## 3. Problem Definition

The objective of FSS is to segment objects from arbitrary classes, using few annotated support samples of each class. The training and testing datasets, denoted as $\mathcal{D}_{\text{train}}$ and $\mathcal{D}_{\text{test}}$, contain disjoint sets of classes, respectively represented as $\mathcal{C}_{\text{base}}$ and $\mathcal{C}_{\text{novel}}$, where $\mathcal{C}_{\text{base}} \cap \mathcal{C}_{\text{novel}} = \emptyset$. This ensures that no classes in $\mathcal{D}_{\text{train}}$ appear in $\mathcal{D}_{\text{test}}$, making FSS a challenging problem of generalizing to unseen classes. To achieve this, FSS employs episodic training, where both $\mathcal{D}_{\text{train}}$ and $\mathcal{D}_{\text{test}}$ are divided into a series of episodes. Each episode in a $k$-shot setting comprises a support set $(I_S^n, M_S^n)_{n=1}^k$ and a query set $(I_Q, M_Q)$, all related to a particular target class. Here, $I_S^n$ and $M_S^n$ denote the $n$-th support image and its binary mask, while $I_Q$ and $M_Q$ represent the query image and its corresponding mask. During training, the model learns to use the support set $\mathcal{S}$ to guide segmentation on $I_Q$ for classes in $\mathcal{C}_{\text{base}}$. In testing, the model applies this learned segmentation pattern to previously unseen classes in $\mathcal{C}_{\text{novel}}$.

## 4. Methodology

As shown in Figure 2, we design FSSAM to use SAM 2's excellent matching ability to facilitate the segmentation of query FGs. Firstly, query and support images $I_Q$ and $I_S$ are forwarded to image encoder to extract features $F_Q$ and $F_S$. Meanwhile, they are forwarded to Pseudo Prompt Generator (Section 4.1) to obtain FG and discriminative (Disc) prior masks $P_Q^{FG}$ and $P_Q^{Disc}$. Then, support features $F_S$ and support mask prompt $M_S$ are encoded as support memory $Mem_S$ via memory encoder. Similarly, $P_Q^{FG}$ and $P_Q^{Disc}$

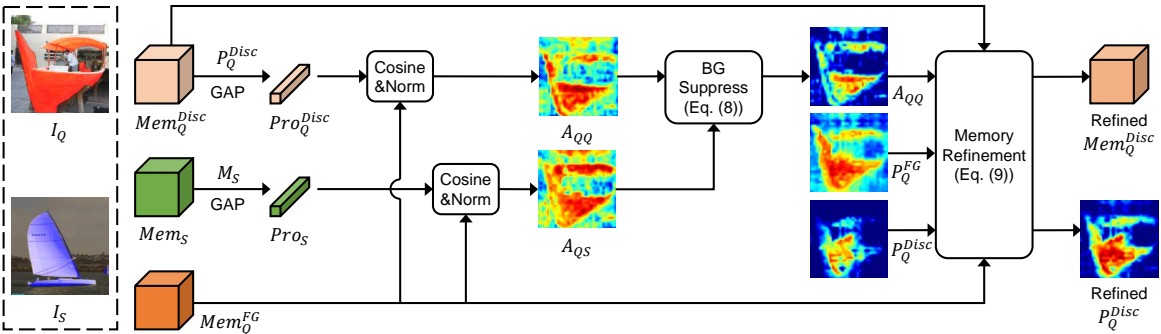

*Figure 3.* Details of Iterative Memory Refinement (IMR). IMR refines Disc memory $Mem_Q^{Disc}$, by measuring its similarity with memory $Mem_Q^{FG}$, and incorporating sufficient query FG features from the latter into the former. Meanwhile, support memory $Mem_S$ (only FG) is used to prevent from fusing many BG features from $Mem_S^{FG}$, so the refined Disc memory $Mem_Q^{Disc}$ can have more complete FG, while still include few BG features. IMR supports iterative refinement.

are used to encode query features $F_Q$ as pseudo query memories $Mem_Q^{FG}$ and $Mem_Q^{Disc}$. After that, Iterative Memory Refinement (Section 4.2) iteratively refines $Mem_Q^{Disc}$, complementing FG features. As $Mem_Q^{Disc}$ always contains BG features, we devise a Support-Calibrated Memory Attention (Section 4.3) to suppress the unexpected BG features in $Mem_Q^{Disc}$ during memory attention, thus, query features $F_Q$ can be appropriately fused as $\hat{F}_Q$, decoded by mask decoder to obtain the final predictions $\hat{M}_Q$.

### 4.1. Pseudo Prompt Generator

**Motivation.** Recall that the straightforward way to adapt SAM 2 for FSS is to take support image $I_S$ as the first frame, use the annotated mask $M_S$ to encode its features $F_S$ as support memory $Mem_S$, which is then propagated into the second frame, i.e., query features $F_Q$, via memory attention. Nevertheless, SAM 2 is trained on abundant video data, where the FG objects in different frames are always the same. Instead, the FG objects in FSS always correspond to visually-different identities, denoted as *intra-class gap* (Fan et al., 2022; Xu et al., 2023). Therefore, SAM 2's FG-FG matching ability is incompatible with that of FSS. Inspired by the FG locating ability of prior masks (Tian et al., 2020; Xu et al., 2023; 2024a), we design Pseudo Prompt Generator (PPG) to alternatively encode query features $F_Q$ as pseudo query memories, so the FG-FG matching and fusion can be compatible with the trained ones.

**Pseudo Prompt Generation.** As shown in Figure 2, we follow existing works (Liu et al., 2023b; Zhang et al., 2024; Chang et al., 2024) to deploy DINOv2 (Oquab et al., 2023), a powerful vision foundation model trained with discriminative self-supervised learning paradigm, to generate discriminative query and support features as:

$$F_Q^{DINO} = \text{DINOv2}(I_Q), F_S^{DINO} = \text{DINOv2}(I_S) \quad (1)$$

Then, we follow AENet (Xu et al., 2024a) to compress $F_S^{DINO}$ into FG and BG prototypes via global average pooling. After that, two normalized cosine similarities, $P_Q^{FG}$ and $P_Q^{BG}$, are measured between prototypes and $F_Q^{DINO}$. Finally, Disc prior mask $P_Q^{Disc}$ is obtained via a subtraction from $P_Q^{FG}$ to $P_Q^{BG}$, and the negative values are set as 0:

$$Pro_S^* = \text{GAP}(F_S^{DINO}, M_S^*) \quad (2)$$

$$P_Q^* = \text{Reshape}(\text{Norm}(\text{Cos}(F_Q^{DINO}, Pro_S^*))) \quad (3)$$

$$P_Q^{Disc} = \text{Norm}(\text{ReLU}(P_Q^{FG} - P_Q^{BG})) \quad (4)$$

where $* \in \{FG, BG\}$, $\text{GAP}(\cdot)$ means global average pooling, $\text{Norm}(\cdot)$ is min-max normalization. $P_Q^{FG}$ and $P_Q^{BG}$ show the probabilities of a query pixel being FG or BG, and $P_Q^{Disc}$ means whether a pixel is more likely to be FG than BG, appearing to be more discriminative prior masks. Some examples are included in Figure 5(b) and Appendix D.2.1.

**Memory Encoding.** After obtaining prior masks, query and support features $F_Q$ and $F_S$ will be encoded as memories:

$$\begin{aligned} Mem_S &= \text{ME}(F_S, M_S) \\ Mem_Q^{FG} &= \text{ME}(F_Q, P_Q^{FG}) \\ Mem_Q^{Disc} &= \text{ME}(F_Q, P_Q^{Disc}) \end{aligned} \quad (5)$$

where $\text{ME}(\cdot)$ is SAM 2's memory encoder.

**Extension to $k$-shot.** Each support sample is used to generate a pair of FG and Disc prior masks, averaged as $P_Q^{FG}$ and $P_Q^{Disc}$. Besides, $k$ support samples will be encoded into $k$ support memories $Mem_S$ accordingly.

### 4.2. Iterative Memory Refinement

**Motivation.** Both $P_Q^{FG}$ and $P_Q^{Disc}$ have been encoded into pseudo query memories, however, as shown in Figure 3,

$P_Q^{FG}$ always has more complete FG but activates *more BG* regions, while $P_Q^{Disc}$ always has *less complete FG* yet activates less BG regions. Note that (1) if there are more complete FG features in memory, the FG-FG matching in memory attention will naturally be easier; (2) SAM 2's matching ability is quite strong, so the unexpected BG features can easily lead to wrong segmentation of BG objects. As $P_Q^{Disc}$ has much less BG than $P_Q^{FG}$, we take $Mem_Q^{Disc}$ as the main memory, and selectively fuse extra FG features from $Mem_Q^{FG}$. During this process, $Mem_S$ is used to restrain the incorporation of redundant BG features.

**Memory Refinement.** As shown in Figure 3, we first compress $Mem_Q^{Disc}$ and $Mem_S$ into prototypes $Pro_Q^{Disc}$ and $Pro_S$. Then, we measure the cosine similarities between prototypes and $Mem_Q^{FG}$ as:

$$Pro_Q^{Disc} = \text{GAP}(Mem_Q^{Disc}, P_Q^{Disc})$$
$$Pro_S = \text{GAP}(Mem_S, M_S) \tag{6}$$

$$A_{QQ} = \text{Reshape}(\text{Norm}(\text{Cos}(Mem_Q^{FG}, Pro_Q^{Disc})))$$
$$A_{QS} = \text{Reshape}(\text{Norm}(\text{Cos}(Mem_Q^{FG}, Pro_S))) \tag{7}$$

where $A_{QQ} \in \mathbb{R}^{H \times W}$ and $A_{QS} \in \mathbb{R}^{H \times W}$ represent the overall similarities of $Mem_Q^{Disc}$ and $Mem_S$ to $Mem_Q^{FG}$.

As there exist BG features in $Mem_Q^{Disc}$, $A_{QQ}$ appears to have relatively larger similarities to BG (e.g., non-ship) regions, i.e., directly taking this score to refine $Mem_Q^{Disc}$ will take in too many unexpected BG features. As $Mem_S$ contains pure FG features, we can observe $A_{QS}$ has relatively smaller scores on query BG regions. Therefore, we further propose a **BG Suppress** mechanism to filter $A_{QQ}$, whose essence is preserving those regions that are similar to both $Mem_Q^{Disc}$ (FG&BG) and $Mem_S$ (only FG):

$$A_{QQ} = \text{ReLU}(A_{QQ} + (A_{QS} - 1)) \tag{8}$$

where $A_{QQ} \in [0, 1]^{H \times W}$ is regarded as the weight to fuse (1) $Mem_Q^{Disc}$ and $Mem_Q^{FG}$, (2) $P_Q^{Disc}$ and $P_Q^{FG}$:

$$Mem_Q^{Disc} = A_{QQ} \cdot Mem_Q^{FG} + (1 - A_{QQ}) \cdot Mem_Q^{Disc}$$
$$P_Q^{Disc} = A_{QQ} \cdot P_Q^{FG} + (1 - A_{QQ}) \cdot P_Q^{Disc} \tag{9}$$

where $\cdot$ is element-wise multiplication. In Figure 3, the refined prior mask $P_Q^{Disc}$ (1) has more complete FG regions than $P_Q^{Disc}$, and (2) has less BG regions than $P_Q^{FG}$, i.e., the pseudo query prompts and memories get better.

**Iterative Mechanism.** We further propose to iteratively refine $Mem_Q^{Disc}$ and $P_Q^{Disc}$ to incorporate more FG features from $Mem_Q^{FG}$, which can be expressed as:

$$Mem_Q^{Disc}, P_Q^{Disc} = \text{MR}(Mem_Q^{Disc}, P_Q^{Disc},$$
$$Mem_Q^{FG}, Mem_S, M_S, n) \tag{10}$$

where $\text{MR}(\cdot)$ includes Equation (6) to Equation (9), and $n$ is iteration times. In general, with the increase of $n$, the refined memory will have more complete FG but *more BG* features. Some visual impacts are depicted in Figure 5(c).

**Extension to $k$-shot.** Each of $k$ support memories will calculate a similarity, averaged as $A_{QS}$.

**Memory Complexity.** As cosine similarities are measured between prototypes (each with 1 pixel) and FG memory ($N = HW$ pixels), the cost is $\mathcal{O}(n(k+1)N)$ under $k$-shot setting, refine $n$ times, where $n \ll N$ and $k \ll N$.

## 4.3. Support-Calibrated Memory Attention

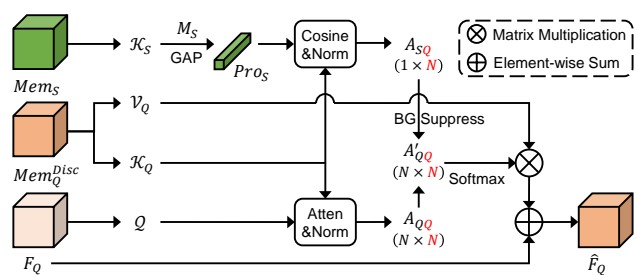

*Figure 4.* Illustrations of Support-Calibrated Memory Attention (SCMA). We only show cross attention in this figure. When performing cross attention between $F_Q$ and $Mem_Q^{Disc}$ (FG&BG), the irrelevant memory (BG) will be suppressed by $Mem_S$ (FG).

**Motivation.** Remind that our goal is to refine $Mem_Q^{Disc}$, trying to (1) make its FG features more complete, and (2) mitigate the side-effects of entangled BG features. In Section 4.2, IMR only focuses on the first point, with the second point untouched, i.e., *IMR cannot drop initially-included BG features but might fuse more BG features into Disc memory*. Hence, we further devise a Support-Calibrated Memory Attention (SCMA) to address this issue. The main idea is preventing query features $F_Q$ from fusing those memory features that are less likely to be FG.

**Support-Calibrated Cross Attention.** As shown in Figure 4, query features $F_Q$ will be projected as $\mathcal{Q} \in \mathbb{R}^{N \times C}$, where $N = HW$ and $C$ is hidden dimension. Similarly, $Mem_S$ is projected as $\mathcal{K}_S \in \mathbb{R}^{N \times C}$, and $Mem_Q^{Disc}$ is projected as $\mathcal{K}_Q \in \mathbb{R}^{N \times C}$ and $\mathcal{V}_Q \in \mathbb{R}^{N \times C}$ for feature fusion:

$$\mathcal{Q} = \text{Proj}(F_Q, \Theta_{\mathcal{Q}}), \qquad \mathcal{V}_Q = \text{Proj}(Mem_Q^{Disc}, \Theta_{\mathcal{V}})$$
$$\mathcal{K}_S = \text{Proj}(Mem_S, \Theta_{\mathcal{K}}), \quad \mathcal{K}_Q = \text{Proj}(Mem_Q^{Disc}, \Theta_{\mathcal{K}}) \tag{11}$$

where $\Theta$ represent the parameters for projection. Note that $\mathcal{K}_Q$ and $\mathcal{K}_S$ share the same projection parameters $\Theta_{\mathcal{K}}$.

Following SAM 2, standard cross attention (i.e., scaled dot product) is conducted between $\mathcal{K}_Q$ and $\mathcal{Q}$, to obtain the attention score $A_{QQ} \in \mathbb{R}^{N \times N}$. Then, a global average

*Table 1.* Quantitative comparisons with state-of-the-arts on PASCAL-$5^i$. "$5^i$" denotes the mIoU score of the $i$-th fold, "Mean" is the averaged mIoU score of 4 folds, "FB-IoU" is averaged from 4 folds. **Bold** values show the best performance.

| Method | 1-shot | | | | | | 5-shot | | | | | |
| --- | --- | --- | --- | --- | --- | --- | --- | --- | --- | --- | --- | --- |
| | $5^0$ | $5^1$ | $5^2$ | $5^3$ | Mean | FB-IoU | $5^0$ | $5^1$ | $5^2$ | $5^3$ | Mean | FB-IoU |
| *Classical FSS* | | | | | | | | | | | | |
| ABCNet (CVPR'23) (Wang et al., 2023b) | 68.8 | 73.4 | 62.3 | 59.5 | 66.0 | 76.0 | 71.7 | 74.2 | 65.4 | 67.0 | 69.6 | 80.0 |
| SCCAN (ICCV'23) (Xu et al., 2023) | 68.3 | 72.5 | 66.8 | 59.8 | 66.8 | 77.7 | 72.3 | 74.1 | 69.1 | 65.6 | 70.3 | 81.8 |
| RiFeNet (AAAI'24) (Bao et al., 2024) | 68.4 | 73.5 | 67.1 | 59.4 | 67.1 | - | 70.0 | 74.7 | 69.4 | 64.2 | 69.6 | - |
| MIANet (CVPR'23) (Yang et al., 2023) | 68.5 | 75.8 | 67.5 | 63.2 | 68.7 | 79.5 | 70.2 | 77.4 | 70.0 | 68.8 | 71.6 | 82.2 |
| HDMNet (CVPR'23) (Peng et al., 2023) | 71.0 | 75.4 | 68.9 | 62.1 | 69.4 | - | 71.3 | 76.2 | 71.3 | 68.5 | 71.8 | - |
| PAM (AAAI'24) (Wang et al., 2024a) | 71.1 | 75.5 | 67.0 | 64.5 | 69.5 | - | 74.7 | 78.0 | 75.3 | 70.8 | 74.7 | - |
| AENet (ECCV'24) (Xu et al., 2024a) | 72.2 | 75.5 | 68.5 | 63.1 | 69.8 | 80.8 | 74.2 | 76.5 | 74.8 | 70.6 | 74.1 | 84.5 |
| AMNet (NIPS'23) (Wang et al., 2023a) | 71.1 | 75.9 | 69.7 | 63.7 | 70.1 | - | 73.2 | 77.8 | 73.2 | 68.7 | 73.2 | - |
| HMNet (NIPS'24) (Xu et al., 2024b) | 72.2 | 75.4 | 70.0 | 63.9 | 70.4 | 81.6 | 74.0 | 77.2 | 74.1 | 70.5 | 73.9 | 84.4 |
| *Foundation-based FSS* | | | | | | | | | | | | |
| Matcher (ICLR'24) (Liu et al., 2023b) | 67.7 | 70.7 | 66.9 | 67.0 | 68.1 | - | 71.4 | 77.5 | 74.1 | 72.8 | 74.0 | - |
| VRP-SAM (CVPR'24) (Sun et al., 2024) | 73.9 | 78.3 | 70.6 | 65.1 | 71.9 | - | - | - | - | - | - | - |
| GF-SAM (NIPS'24) (Zhang et al., 2024) | 71.1 | 75.7 | 69.2 | 73.3 | 72.1 | - | 81.5 | 86.3 | 79.7 | 82.9 | 82.6 | - |
| FounFSS (Arxiv'24) (Chang et al., 2024) | 76.5 | 81.3 | 72.1 | **77.4** | 76.8 | - | 79.5 | 84.8 | 75.8 | 82.5 | 80.7 | - |
| FSSAM (Ours) | **81.6** | **84.9** | **81.6** | 76.0 | **81.0** | **89.4** | **84.1** | **88.5** | **83.8** | **85.0** | **85.4** | **91.9** |

pooling operation is carried out on $Mem_S$ to compress its FG memory features into $Pros_S \in \mathbb{R}^{1 \times C}$. As the FG objects in query and support images are different, and cosine similarity is a much looser similarity operator (Xu et al., 2023) than scaled dot product, we calculate cosine similarity between $Pros_S$ and $\mathcal{K}_Q$ to encourage better FG-FG matching, and obtain similarity score $A_{SQ} \in [-1, 1]^{1 \times N}$.

$$A_{QQ} = \frac{\mathcal{Q} \otimes \mathcal{K}_Q}{\sqrt{d_\mathcal{K}}}, A_{SQ} = \frac{Pros_S \otimes \mathcal{K}_Q}{\|Pros_S\| \otimes \|\mathcal{K}_Q\| + \epsilon} \quad (12)$$

where $\otimes$ means matrix multiplication, $d_\mathcal{K}$ denotes the hidden dimension of $\mathcal{K}$, and $\epsilon$ is a small constant to avoid 0. Next, both $A_{QQ}$ and $A_{SQ}$ will be normalized into $[0, 1]$, **BG Suppress** is then performed to suppress the attention scores of irrelevant BG pixels in $A_{QQ}$ as:

$$A'_{QQ} = \text{Norm}(A_{QQ}) + (\text{Norm}(A_{SQ}) - 1)$$
$$A_{QQ} = A_{QQ} + \alpha \cdot \bar{A}'_{QQ} \quad (13)$$

where $\alpha$ is a scaling factor and is empirically set as 10, $\bar{A}'_{QQ}$ means preserving those entries with negative values. Then, $A_{QQ}$ is normalized by softmax and used to aggregate features from $Mem_Q^{Disc}$ and projected as $\hat{F}_Q$, which will be fused with input features $F_Q$ via skip connection:

$$\hat{F}_Q = F_Q + \text{Proj}(\text{Softmax}(A_{QQ}) \otimes \mathcal{V}_Q, \Theta_{Out}) \quad (14)$$

where $\Theta_{Out}$ is used for output projection. The visual impacts of SCMA is shown in Figure 6.

**Extension to $k$-shot.** The $k$ support memories will be used to measure $k$ cosine similarities, and averaged to be $A_{SQ}$.

**Memory Complexity.** Remind that the memory cost of original cross attention has already been $\mathcal{O}(N^2)$, and the extra cost of our design is $\mathcal{O}(kN)$ under $k$-shot setting.

## 5. Experiments

### 5.1. Experiment Setup

**Datasets.** We evaluate FSSAM on two benchmarks, including PASCAL-$5^i$ (Shaban et al., 2017) and COCO-$20^i$ (Nguyen & Todorovic, 2019). PASCAL-$5^i$ includes 20 classes, built upon the PASCAL VOC 2012 (Everingham et al., 2010) with additional annotations from SDS (Hariharan et al., 2014). COCO-$20^i$, derived from the MSCOCO (Lin et al., 2014), is more challenging, comprising 80 classes. Both datasets are divided into 4 disjoint folds for cross-validation, with each fold containing 5 classes for PASCAL-$5^i$ and 20 classes for COCO-$20^i$. In each iteration, 3 folds are used for training, and the remaining fold is used for testing. Following existing works, we randomly sample 1,000 episodes for testing by default, while the results for error bars evaluation and different number of testing episodes are included in Appendix C.1 and Appendix C.2.

**Evaluation Metrics.** Mean intersection over union (mIoU) and foreground-background IoU (FB-IoU) are utilized.

**Implementation Details.** Please refer to Appendix B.1.

### 5.2. Comparisons with State-of-the-Arts

The quantitative comparisons between our FSSAM and previous state-of-the-arts are shown in Table 1 and Table 2 for PASCAL-$5^i$ and COCO-$20^i$. Specifically, we select some classical FSS methods, as well as more recent foundation-based methods for comparisons. Following existing works, the comparisons are conducted under both 1-shot and 5-shot settings. We could draw the following conclusions from the tables: (1) Foundation-based methods consistently behave better than classical methods, showing the necessity of introducing foundation models to take over some of FSS models' duties, e.g., the matching ability can thus be less likely to

*Table 2.* Quantitative comparisons with state-of-the-arts on COCO-$20^i$. "$20^i$" denotes the mIoU score of the $i$-th fold, "Mean" is the averaged mIoU score of 4 folds, "FB-IoU" is averaged from 4 folds. **Bold** values show the best performance.

| Method | 1-shot | | | | | | 5-shot | | | | | |
|---|---|---|---|---|---|---|---|---|---|---|---|---|
| | $20^0$ | $20^1$ | $20^2$ | $20^3$ | Mean | FB-IoU | $20^0$ | $20^1$ | $20^2$ | $20^3$ | Mean | FB-IoU |
| *Classical FSS* | | | | | | | | | | | | |
| ABCNet (CVPR'23) (Wang et al., 2023b) | 42.3 | 46.2 | 46.0 | 42.0 | 44.1 | 69.9 | 45.5 | 51.7 | 52.6 | 46.4 | 49.1 | 72.7 |
| RiFeNet (AAAI'24) (Bao et al., 2024) | 39.1 | 47.2 | 44.6 | 45.4 | 44.1 | - | 44.3 | 52.4 | 49.3 | 48.4 | 48.6 | - |
| SCCAN (ICCV'23) (Xu et al., 2023) | 40.4 | 49.7 | 49.6 | 45.6 | 46.3 | 69.9 | 47.2 | 57.2 | 59.2 | 52.1 | 53.9 | 74.2 |
| MIANet (CVPR'23) (Yang et al., 2023) | 42.5 | 53.0 | 47.8 | 47.4 | 47.7 | 71.5 | 45.8 | 58.2 | 51.3 | 51.9 | 51.7 | 73.1 |
| PAM (AAAI'24) (Wang et al., 2024a) | 44.1 | 55.0 | 46.5 | 48.5 | 48.5 | - | 48.1 | 60.8 | 54.8 | 51.9 | 53.9 | - |
| AENet (ECCV'24) (Xu et al., 2024a) | 43.1 | 56.0 | 50.3 | 48.4 | 49.4 | 73.6 | 51.7 | 61.9 | 57.9 | 55.3 | 56.7 | 76.5 |
| HDMNet (CVPR'23) (Peng et al., 2023) | 43.8 | 55.3 | 51.6 | 49.4 | 50.0 | 72.2 | 50.6 | 61.6 | 55.7 | 56.0 | 56.0 | 77.7 |
| AMNet (NIPS'23) (Wang et al., 2023a) | 44.9 | 55.8 | 52.7 | 50.6 | 51.0 | 72.9 | 52.0 | 61.9 | 57.4 | 57.9 | 57.3 | 78.8 |
| HMNet (NIPS'24) (Xu et al., 2024b) | 45.5 | 58.7 | 52.9 | 51.4 | 52.1 | 74.5 | 53.4 | 64.6 | 60.8 | 56.8 | 58.9 | 77.6 |
| *Foundation-based FSS* | | | | | | | | | | | | |
| Matcher (ICLR'24) (Liu et al., 2023b) | 52.7 | 53.5 | 52.6 | 52.1 | 52.7 | - | 60.1 | 62.7 | 60.9 | 59.2 | 60.7 | - |
| FounFSS (Arxiv'24) (Chang et al., 2024) | 56.0 | 61.3 | 57.9 | 58.8 | 58.5 | - | 61.4 | 69.4 | 65.9 | 64.9 | 65.4 | - |
| GF-SAM (NIPS'24) (Zhang et al., 2024) | 56.6 | 61.4 | 59.6 | 57.1 | 58.7 | - | 67.1 | 69.4 | **66.0** | 64.8 | 66.8 | - |
| VRP-SAM (CVPR'24) (Sun et al., 2024) | 56.8 | 61.0 | **64.2** | 59.7 | 60.4 | - | - | - | - | - | - | - |
| FSSAM (Ours) | **59.9** | **65.6** | 62.1 | **61.6** | **62.3** | **77.3** | **68.6** | **74.0** | 64.5 | **69.9** | **69.3** | **82.9** |

*Table 3.* Component-wise ablation study. "PPG", "IMR" and "SCMA" are Pseudo Prompt Generator, Iterative Memory Refinement and Support-Calibrated Memory Attention, respectively.

| PPG | IMR | SCMA | $5^0$ | $5^1$ | $5^2$ | $5^3$ | Mean | FB-IoU |
|---|---|---|---|---|---|---|---|---|
| | | | 71.8 | 74.4 | 71.6 | 59.9 | 69.4 | 80.3 |
| ✓ | | | 79.0 | 82.1 | 78.4 | 73.2 | 78.2 | 87.3 |
| ✓ | ✓ | | 80.7 | 84.1 | 79.4 | 75.2 | 79.6 | 88.3 |
| ✓ | | ✓ | 80.0 | 82.8 | 79.2 | 74.1 | 79.0 | 87.8 |
| ✓ | ✓ | ✓ | **81.6** | **74.9** | **81.6** | **76.0** | **81.0** | **89.4** |

overfit the base classes. Besides, their performance gap appears to be larger on COCO-$20^i$ dataset, and we attribute this to the fact that the samples of COCO-$20^i$ are much more complicated than those of PASCAL-$5^i$, including tiny objects, complex BG, etc. Therefore, models need to learn more complex knowledge to handle them, while the risk of overfitting also increases. (2) FSSAM can surpass other baselines by large margins, setting new state-of-the-arts. For example, the 1-shot mIoU score is 4.2% better than that of the best baseline FounFSS, and the FB-IoU score can reach 89.4%. Under 5-shot setting, the gap is more prominent (e.g., 85.4% v.s. 80.7%), demonstrating its effectiveness.

### 5.3. Ablation Study

If not explicitly specified, the experiments are conducted on PASCAL-$5^i$, under 1-shot setting. Kindly remind that more experiments are included in Appendix.

**Component-wise Ablation Study.** To validate the effectiveness of each design, the detailed component-wise ablation studies are provided in Table 3. We start from pure SAM 2-S (46 M) model with the most straightforward idea (Figure 1(b) and Appendix D.1), and the averaged mIoU score is 69.4%, which cannot surpass some classical FSS methods. As explained earlier, we attribute this to the existence of

incompatible matching. When we deploy PPG to enable compatible same-objects matching, the score can be greatly improved from 69.4% to 78.2%, showing the reasonability of our main idea. Nevertheless, the generated prior masks always have less FG yet more BG regions than the real masks. When we use IMR to introduce more query FG features into the memory, the mIoU can be further improved by 1.4%. Meanwhile, if we deploy SCMA to suppress the unexpected BG features during cross attention, the score is 79.0%. Once IMR and SCMA are both utilized, the final score can reach 81.0%, setting new state-of-the-arts.

**Qualitative Results and Pseudo Prompts.** To have a clearer understanding of FSSAM, we jointly visualize some qualitative results of PPG, IMR and final predictions in Figure 5, where: (1) In (b), PPG can generate reasonable FG and Disc priors, coarsely locating the target FG objects, but they each has some problems. For example, FG priors contain much more BG regions, as SAM 2 has quite strong matching ability, the obtained predictions are likely to have unexpected BG regions. As for Disc priors, although there are much less activated BG, the FG regions also become less. (2) In (c), IMR can help to incorporate FG regions from FG prior to Disc prior, struggling to fuse less BG regions. Take the second row as an example, the orange rectangle includes some non-cow animals, which are wrongly considered as cows in FG prior. During IMR, Disc prior gradually activate more FG regions, while the orange box keeps correctly inactivated, leading to correct predictions.

**Parameter Study on IMR.** We vary the iteration times $n$ of IMR in the final model, and show their effects in Figure 5(c) and Table 4. It can observed: (1) with the increase of $n$, the refined Disc memory and prior mask can be supplemented with more FG features and regions, (2) if they are initially entangled with BG, IMR cannot help to filter them, and (3)

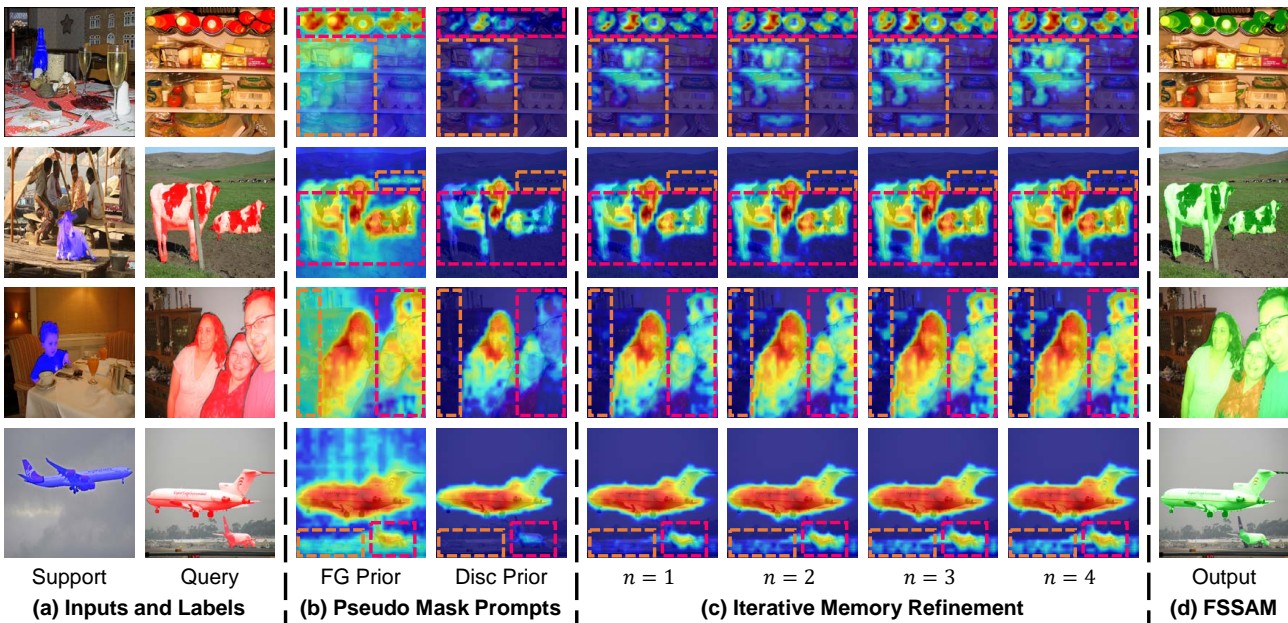

| Support | Query | FG Prior | Disc Prior | $n = 1$ | $n = 2$ | $n = 3$ | $n = 4$ | Output |
|---------|-------|----------|------------|---------|---------|---------|---------|--------|
| **(a) Inputs and Labels** | | **(b) Pseudo Mask Prompts** | | **(c) Iterative Memory Refinement** | | | | **(d) FSSAM** |

*Figure 5.* Qualitative illustrations of (a) query and support samples, (b) pseudo priors (mask prompts), (c) iterative memory refinement, and (d) outputs. We plot some rectangles to highlight some FG and BG areas. In (b), FG prior appears to have more complete FG but *more wrongly activated BG* regions, while BG prior has *less complete FG* yet less wrongly activated BG regions. In (c), the FG regions of FG prior can be propagated to Disc prior. With more iterations, more FGs are fused into Disc prior, but also fused with *more BG* regions.

*Table 4.* Parameter study on Iterative Memory Refinement (IMR).

| #Iterations | $5^0$ | $5^1$ | $5^2$ | $5^3$ | Mean | FB-IoU |
|-------------|-------|-------|-------|-------|------|--------|
| 1 | 80.5 | 83.4 | 79.2 | 73.9 | 79.3 | 88.2 |
| 2 | 81.2 | 84.4 | 80.4 | 75.1 | 80.3 | 88.9 |
| 3 | **81.6** | **84.9** | **81.6** | 76.0 | **81.0** | **89.4** |
| 4 | 81.5 | 84.6 | 81.0 | **76.3** | 80.8 | 89.0 |

some BG regions of FG prior might also be propagated into Disc prior instead. Therefore, trade-offs should be made between introducing more query FG and more query BG features. Table 4 demonstrates that using more iterations can boost the performance at first, e.g., the mIoU score is improved from 79.3% to 81.0%, yet further refinement cannot improve the score anymore, as the incorporation of more BGs hinders the FG-FG matching and fusion. Hence, we set iteration times $n$ as 3 in this paper.

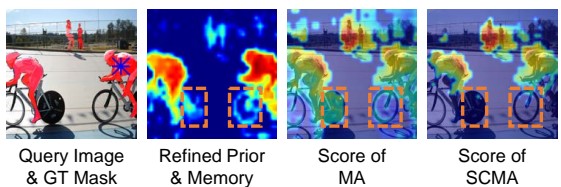

| Query Image | Refined Prior | Score of | Score of |
| & GT Mask | & Memory | MA | SCMA |

*Figure 6.* Visual impacts of SCMA. The last two columns are the cross attention scores of the blue star. Orange rectangles represent some challenging areas that should be classified as BG.

**Visualizations of SCMA.** An example is depicted in Figure 6, where the second column represent the refined query pseudo memory (prior) of IMR, the last two columns represent the cross attention scores between the sampled query FG pixel and the pseudo memory. We could observe: (1) The refined memory usually contain unexpected BG features, e.g., wheels. (2) The original memory attention (MA) cannot filter these unexpected BGs, leading to the wrong segmentation. (3) Our designed SCMA can suppress the attention scores of these positions, with the help of support memory. Kindly note that SCMA can reduce the scores between query FG pixels and unexpected BG pixels (in memory) by 41.5%, the details are included in Appendix C.4.

## 6. Conclusion

In this paper, we incorporate SAM 2 into FSS to leverage its well-learned FG-FG matching ability. The simple way is encoding support features as memory, used to match and enhance query features. However, the matching of SAM 2 belongs to same-objects matching, while the FG objects in query and support are different. Therefore, we design a PPG to generate pseudo query memories, making such matching compatible. Furthermore, we design IMR to supplement this memory with more query FG features, and devise a SCMA to mitigate the side-effects of unexpected BG features in pseudo memory. Extensive experiments have been conducted to validate the effectiveness of our design.

## Acknowledgements

This study is supported under the RIE2020 Industry Alignment Fund – Industry Collaboration Projects (IAF-ICP) Funding Initiative, as well as cash and in-kind contribution from the industry partner(s).

## Impact Statement

This paper presents work whose goal is to advance the field of Machine Learning. There are many potential societal consequences of our work, none which we feel must be specifically highlighted here.

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

# A. Code

The source code is available at our Github repository https://github.com/Sam1224/FSSAM.

# B. Additional Details

## B.1. Implementation Details

All models are trained with 4 NVIDIA V100 (32G) GPUs, and tested with 1 V100 GPU. Remind that our FSSAM is built upon SAM 2, without introducing additional learnable parameters. We deploy AdamW to optimize SAM 2's memory encoder, memory attention and mask decoder, with other components frozen. The learning rate is initialized as 0.001, and decayed with a polynomial scheduler (Tian et al., 2020). We follow SCCAN (Xu et al., 2023) to perform data augmentation, and adopt Dice loss (Milletari et al., 2016) for training (fine-tuning). During training, the images are randomly cropped as $512 \times 512$ patches, and the testing predictions will be resized back to the original shape for metric calculation. The batch size is set as 8, with 2 samples distributed to each GPU. Following existing works (Wang et al., 2023a), the training epochs are set as 300 and 75 for PASCAL-$5^i$ and COCO-$20^i$. We employ SAM 2-S (46 M) and DINOv2-B (86 M) for the main experiments, and study different model sizes in Appendix C.3.

# C. Additional Experiments

In this section, we conduct experiments to validate the effectiveness of FSSAM, including error bars evaluation (Appendix C.1), the performance and efficiency with different testing episodes (Appendix C.2), different sizes of SAM 2 and DINOv2 (Appendix C.3), study the quantitative impacts of SCMA (Appendix C.4), provide more quantitative comparisons (Appendix C.5), emphasize the necessity of fine-tuning (Appendix C.6), and compare our parameter number with baselines (Appendix C.7).

## C.1. Error Bars Evaluation

We conduct error bars evaluation in Table 5 to examine the robustness towards randomness. The random seeds are selected from {321 (default), 0, 1, 2, 3}, and each seed would be responsible for sampling 1,000 testing episodes from PASCAL-$5^i$ and COCO-$20^i$, i.e., the query and support samples will vary, with the altering of random seeds. It could be observed from the table: (1) In PASCAL-$5^i$, the mIoU and FB-IoU scores, averaged from 5 random seeds, are 81.1±0.5 and 89.3±0.3, similar to the reported values (in main paper) 81.0 and 89.3, when the seed is set as 321. This could demonstrate that our proposed FSSAM is robust; (2) COCO-$20^i$ is much more challenging than PASCAL-$5^i$, e.g., it includes multiple objects, small objects, complex

*Table 5.* Error bars evaluation on PASCAL-$5^i$ and COCO-$20^i$ datasets. The number of testing episodes is 1,000. The selected random seeds are {0, 1, 2, 3, 321}. "Mean" is the averaged mIoU score of 4 folds, "F-i" means the i-th fold.

| Dataset | Seed | F-0 | F-1 | F-2 | F-3 | Mean | FB-IoU |
|---|---|---|---|---|---|---|---|
| PASCAL-$5^i$ | 0 | 81.6 | 85.7 | 79.9 | 77.6 | 81.2 | 89.4 |
| | 1 | 81.4 | 83.3 | 79.9 | 77.3 | 80.5 | 88.8 |
| | 2 | 82.1 | 85.4 | 80.8 | 78.8 | 81.8 | 89.7 |
| | 3 | 81.2 | 84.1 | 79.8 | 78.3 | 80.9 | 89.2 |
| | 321 | 81.6 | 84.9 | 81.6 | 76.0 | 81.0 | 89.4 |
| | Mean | 81.6 | 84.7 | 80.4 | 77.6 | 81.1 | 89.3 |
| | Std | 0.3 | 1.0 | 0.8 | 1.1 | 0.5 | 0.3 |
| COCO-$20^i$ | 0 | 57.5 | 67.1 | 62.3 | 60.5 | 61.9 | 78.2 |
| | 1 | 61.8 | 63.1 | 62.6 | 59.7 | 61.8 | 77.6 |
| | 2 | 55.8 | 63.7 | 58.1 | 63.6 | 60.3 | 78.0 |
| | 3 | 56.0 | 64.8 | 60.3 | 60.4 | 60.4 | 76.9 |
| | 321 | 59.9 | 65.6 | 62.1 | 61.6 | 62.3 | 77.3 |
| | Mean | 58.2 | 64.9 | 61.1 | 61.2 | 61.3 | 77.6 |
| | Std | 2.6 | 1.6 | 1.9 | 1.5 | 0.9 | 0.5 |

background, etc. Therefore, with the change of random seeds, the sampled 1,000 episodes could have quite different testing difficulties, e.g., the mIoU and FB-IoU scores are 61.3±0.9 and 77.6±0.5, where the standard deviation values appear to be consistently larger than those in PASCAL-$5^i$. Hence, we study different episode number next.

## C.2. Different Number of Testing Episodes

*Table 6.* Performance on COCO-$20^i$ with different number of testing episodes in {1,000, 4,000, 10,000, 20,000}. The random seed is fixed as 321. "Mean" is the averaged mIoU score of 4 folds, "$20^i$" denotes the mIoU score of the i-th fold.

| #Test | $20^0$ | $20^1$ | $20^2$ | $20^3$ | Mean | FB-IoU | Time (s) |
|---|---|---|---|---|---|---|---|
| 1,000 | 59.9 | 65.6 | 62.1 | 61.6 | 62.3 | 77.3 | 353.2 |
| 4,000 | 58.6 | 64.1 | 65.3 | 61.5 | 62.4 | 77.6 | 1433.5 |
| 10,000 | 56.7 | 64.8 | 64.6 | 60.6 | 61.7 | 77.5 | 3579.2 |
| 20,000 | 57.2 | 64.6 | 63.1 | 60.9 | 61.5 | 77.4 | 7006.1 |
| Mean | 58.1 | 64.8 | 63.8 | 61.2 | 62.0 | 77.5 | 3093.0 |

As mentioned earlier, COCO-$20^i$ is a challenging dataset, the testing results of 1,000 randomly sampled testing episodes may not be convincing enough. Therefore, we change the number of testing episodes as {1,000, 4,000, 10,000, 20,000}, and present the testing results in Table 6 to study the impacts of different episode number. Meanwhile, we also measure the average time cost for testing each fold. We use SAM 2-S (46 M) and DINOv2-B (86 M) for the experiments. The testings are executed on single NVIDIA V100 with 32 GB onboard memory. From the table, we can draw the following conclusions: (1) Testing with different number of episodes do not have huge impacts on the obtained scores, e.g., the mIoU score of testing 1,000 episodes is 62.3%, while the average mIoU score of testing 1,000, 4,000, 10,000 and 20,000 episodes is 62.0%, where there is

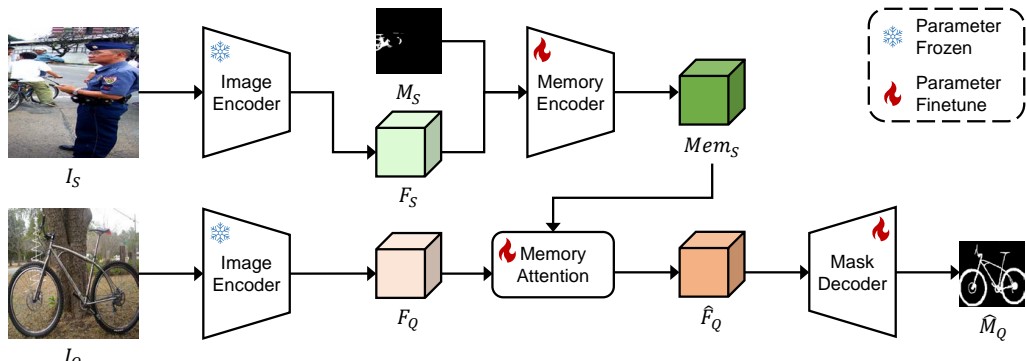

*Figure 7.* Overview of the simple idea to use SAM 2 for FSS. First of all, query and support features $F_Q$ and $F_S$ are extracted by the image encoder. Then, support features $F_S$ will be encoded as support memory $Mem_S$, taking support mask $M_S$ as mask prompt. After that, query features $F_Q$ are matched and fused with support memory $Mem_S$ in memory attention to obtain the fused query features $\hat{F}_Q$. Finally, $\hat{F}_Q$ is decoded by mask decoder to obtain the predictions $\hat{M}_Q$.

merely a gap of 0.3%, showing the stability and superiority of the model; (2) As the deployment of foundation models would introduce more parameters, the amount of calculations would be larger, i.e., the efficiency would be inevitably reduced. According to the table, although we deploy large foundation models (with 46+86=132 M parameters in total), the time cost is still affordable, e.g., the FPS is around 2.8.

### C.3. Different Sizes of SAM 2 and DINOv2

*Table 7.* Study on different model sizes of SAM 2 and DINOv2 on PASCAL-$5^i$. The value in $(\cdot)$ means the number of parameters (M), e.g., for SAM 2, S (46) means SAM 2-S has 46 M parameters. "Mean" is the averaged mIoU score of 4 folds, "$5^i$" denotes the mIoU score of the $i$-th fold.

| SAM 2 | DINOv2 | $5^0$ | $5^1$ | $5^2$ | $5^3$ | Mean | FB-IoU |
|---|---|---|---|---|---|---|---|
| S (46) | B (86) | 81.6 | 84.9 | 81.6 | 76.0 | 81.0 | 89.4 |
| | L (300) | 81.9 | 82.5 | 79.4 | 78.5 | 80.6 | 88.9 |
| B (80.8) | B (86) | 79.8 | 84.8 | 79.0 | 75.8 | 79.9 | 88.3 |
| | L (300) | 80.2 | 82.8 | 77.9 | 78.2 | 79.8 | 88.7 |
| L (224.4) | B (86) | 82.0 | 85.1 | 80.3 | 78.7 | 81.5 | 89.7 |
| | L (300) | 82.1 | 82.6 | 79.1 | 80.6 | 81.1 | 89.5 |

We further study the impacts of SAM 2's and DINOv2's sizes on the performance and obtain the results in Table 7. Specifically, we select 3 out of 4 versions of SAM 2, including small (S), base plus (B) and large (L), comprising 46 M, 80.8 M and 224.4 M parameters, respectively. For DINOv2, we choose base (B) and large (L), which have 86 M and 300 M parameters. According to the table, we could find: (1) It is not the case that the foundation models are the larger, the better. For example, remind that the only responsibility of DINOv2 is to generate query prior masks, acting as pseudo mask prompts to encode pseudo query memories.

When SAM 2-S (46 M) is deployed, upgrading DINOv2 from B (86 M) to L (300 M) cannot boost the performance. Particularly, the scores on fold $5^1$ and $5^2$ are not as good as those with DINOv2-B (86 M). After delving into the visualizations of these two folds, we observe DINOv2-B (86 M) can generate better prior masks for them. Although DINOv2-L (300 M) can improve the score by 1.5% on fold $5^3$, the average mIoU score cannot surpass that of DINOv2-B (86 M), yet the computational cost is much larger (86 M $\rightarrow$ 300 M). (2) When DINOv2 is fixed as B (86 M), upgrading SAM 2 from S (46 M) to L (224.4 M) can witness an mIoU improvement of 0.5%, showing the advantages of introducing more parameters. Nevertheless, the parameter number would be approximately 5 times larger.

After making trade-offs between the performance and the computational cost, we deploy SAM 2-S (46 M) and DINOv2-B (86 M) in this paper.

### C.4. Quantitative Impacts of SCMA

We include a visual example in the main paper, and further quantify the impacts of SCMA in Table 8. Kindly remind that the main idea of SCMA is the **BG Suppress** mechanism, which selectively suppresses the cross attention scores of those unexpected BG pixels (in memory) with a bias.

Note that SCMA is devised from memory attention (MA), and they both comprise 4 attention layers, each of which further includes a pair of self and cross attentions. Self attentions only operate on query features, while cross attentions aim to dynamically match and fuse query features ($\mathcal{Q}$) with the pseudo query memory ($\mathcal{K}\&\mathcal{V}$).

Both query features and the pseudo query memory have the same $N = HW$ pixels. After cross attention, the score has a size of $N \times N$, where the two dimensions refer to

*Table 8.* Quantitative Impacts of SCMA. SCMA includes 4 consecutive attention layers, each has a pair of self and cross attentions. Cross attention is used to match and fuse query features with the pseudo query memory. The values of "MA" and "SCMA" are the (before-softmax) average cross attention scores of each query FG pixel with the unexpected BG pixels in memory. "Mean" is the averaged values of 4 folds, "$5^i$" denotes the mIoU score of the $i$-th fold.

| Layer | $5^0$ | | | $5^1$ | | | $5^2$ | | | $5^3$ | | | Mean | | |
| --- | --- | --- | --- | --- | --- | --- | --- | --- | --- | --- | --- | --- | --- | --- | --- |
| | MA | SCMA | Gap | MA | SCMA | Gap | MA | SCMA | Gap | MA | SCMA | Gap | MA | SCMA | GAP |
| 1 | -5.9 | -8.2 | 38.4% | -4.6 | -6.3 | 36.5% | -5.1 | -7.2 | 40.2% | -4.7 | -6.3 | 33.4% | -5.1 | -7.0 | 37.3% |
| 2 | -4.8 | -7.1 | 48.0% | -4.8 | -6.3 | 32.4% | -5.1 | -6.9 | 36.0% | -4.9 | -6.4 | 30.8% | -4.9 | -6.7 | 36.8% |
| 3 | -4.4 | -7.0 | 60.2% | -4.6 | -6.6 | 43.2% | -3.7 | -6.0 | 61.6% | -3.9 | -6.0 | 52.6% | -4.1 | -6.4 | 54.0% |
| 4 | -6.6 | -9.0 | 37.1% | -4.2 | -6.0 | 42.4% | -4.8 | -7.0 | 43.6% | -4.4 | -6.1 | 38.1% | -5.0 | -7.0 | 40.0% |
| Mean | -5.4 | -7.8 | 44.5% | -4.6 | -6.3 | 38.5% | -4.7 | -6.8 | 44.2% | -4.5 | -6.2 | 38.1% | -4.8 | -6.8 | 41.5% |

*Table 9.* Quantitative comparisons on LVIS-92$^i$. "Mean" is the averaged values of 10 folds, "92$^i$" denotes the mIoU score of the $i$-th fold.

| Method | 1-shot | | | | | | | | | | | |
| --- | --- | --- | --- | --- | --- | --- | --- | --- | --- | --- | --- | --- |
| | $92^0$ | $92^1$ | $92^2$ | $92^3$ | $92^4$ | $92^5$ | $92^6$ | $92^7$ | $92^8$ | $92^9$ | Mean | FB-IoU |
| Matcher (Liu et al., 2023b) | 31.4 | 30.9 | 33.7 | 38.1 | 30.5 | 32.5 | 35.9 | 34.2 | 33.0 | 29.7 | 33.0 | 66.2 |
| SINE (Liu et al., 2024) | 28.3 | 31.0 | 31.9 | 34.6 | 30.0 | 31.9 | 32.2 | 33.7 | 30.6 | 27.8 | 31.2 | 63.5 |
| FSSAM (Ours) | **34.7** | **37.8** | **37.2** | **41.1** | **33.9** | **38.1** | **40.6** | **38.9** | **36.9** | **33.8** | **37.3** | **68.4** |

features and memory, respectively. Then, we use the query ground truth label $M_Q$ to preserve FG pixels in the first dimension as $N_{FG} \times N$. After that, we subtract the query label $M_Q$ from the refined Disc prior mask $P_Q^{Disc}$, and multiply with the second dimension to preserve the entries with positive values, represented as $N_{FG} \times N_{BG}$. This reduced score means the similarities of each query FG pixel to the unexpected BG pixels in the pseudo query memory. The values are the smaller the better, so query FG pixels would not fuse too many unexpected BG features from the memory, leading to more accurate segmentation results.

As presented in Table 8, the provided scores have not been normalized by softmax, because those after softmax will be quite small, being difficult to see the impacts of SCMA. We could observe that the overall average gap between the reduced cross attention scores of MA and SCMA are 41.5%, showing the effectiveness of SCMA to mitigate the side-effects of unexpected BG features in pseudo query memory.

## C.5. More Quantitative Comparisons

*Table 10.* Quantitative comparisons on PASCAL-Part.

| Method | 1-shot | | | | |
| --- | --- | --- | --- | --- | --- |
| | F-0 | F-1 | F-2 | F-3 | Mean |
| Matcher (Liu et al., 2023b) | **37.1** | 32.4 | 33.7 | 38.1 | 42.9 |
| FSSAM (Ours) | 34.7 | **37.8** | **37.2** | **41.1** | **46.4** |

Following Matcher (Liu et al., 2023b), we evaluate the performance of the designed FSSAM on two more challenging datasets, including LVIS-92$^i$ (Liu et al., 2023b) (derived from LVIS (Gupta et al., 2019)) and PASCAL-Part (Morabia

et al., 2020; Liu et al., 2023b) (PASCAL VOC (Everingham et al., 2010) with body part annotations (Chen et al., 2014)), and the results are presented in Table 9 and Table 10, respectively. We select both Matcher and SINE (Liu et al., 2024) for comparisons, where SINE is trained with the whole COCO (80 classes) and directly tested on LVIS-92$^i$. Similarly, we take our trained FSSAM on COCO-20$^0$ (trained with 60 classes, under 1-shot setting) to perform evaluation on both LVIS-92$^i$ and PASCAL-Part. We can observe that our FSSAM can show excellent generalizability (COCO-20$^i \to$ LVIS-92$^i$, and COCO-20$^i \to$ PASCAL-Part).

## C.6. Necessity of Fine-tuning

*Table 11.* Experiment on the necessity of fine-tuning.

| Method | 1-shot | | | | |
| --- | --- | --- | --- | --- | --- |
| | $5^0$ | $5^1$ | $5^2$ | $5^3$ | Mean |
| SAM 2 w/o FT | 49.1 | 43.7 | 51.1 | 35.0 | 44.7 |
| SAM 2 w/ FT | 71.8 | 74.4 | 71.6 | 59.9 | 69.4 |
| FSSAM w/o FT | 61.9 | 59.8 | 61.0 | 51.4 | 58.5 |
| FSSAM w/ FT | 81.6 | 74.9 | 81.6 | 76.0 | 81.0 |

We study whether fine-tuning (FT) is required when using SAM 2 for FSS, and present the results in Table 11. We can observe fine-tuning is necessary: (1) Original SAM 2's memory attention is trained for **same-object matching**, while the matching of FSS is between *different query and support FG objects*, which we call it *incompatible FG-FG matching*. As we can observe from the table, with or without fine-tuning show prominent performance gap; (2) FSSAM can consistently outperform SAM 2 by large margins, since we design modules to resolve the issue. However, FSSAM

still requires fine-tuning, because the pseudo mask prompt is *inaccurate*, covering *incomplete FG regions* and *unexpected BG regions*, which is different from SAM 2's mask prompt.

## C.7. Comparisons on Parameter Number

*Table 12.* Compare with existing baselines in terms of parameter number. "#Total" and "#Learnable" denote the total number of parameters and the learnable parameter number in million (M).

| Method | #Total | #Learnable | mIoU |
|---|---|---|---|
| HDMNet (Peng et al., 2023) | 51 | 4 | 69.4 |
| AMNet (Wang et al., 2023a) | 54 | 7 | 70.1 |
| HMNet (Xu et al., 2024b) | 62 | 15 | 70.4 |
| Matcher (Liu et al., 2023b) | 941 | 0 | 68.1 |
| VRP-SAM (Sun et al., 2024) | 666 | 2 | 71.9 |
| GF-SAM (Zhang et al., 2024) | 941 | 0 | 72.1 |
| FounFSS (Chang et al., 2024) | 87 | 1 | 76.8 |
| FSSAM (Ours) | 132 | 11 | 81 |

To further show the computational burden of FSSAM, we select some methods and summarize their parameter number, learnable parameter number, and the 1-shot mIoU on PASCAL-$5^i$ in Table 12. The first 3 rows refer to classical FSS methods that use ResNet50 as the pretrained backbone, and the remaining methods refer to foundation-based FSS methods that use DINOv2 and/or SAM. For our finalized model, we use DINOv2-B (86M) and SAM 2-S (46M) without extra parameters (kindly remind our proposed modules are parameter-free), and fine-tune part of SAM 2's parameters. It can be observed from the table: (1) Among foundation-based FSS methods, our parameter number is much smaller than most of them, while our performance is consistently much better; (2) Compared to classical FSS methods, though we use more parameters, the difference is not as large as expected, while the performance gap is quite prominent, so we believe the additional cost is worthy.

# D. Additional Figures

We provide details about the simple use of SAM 2 in Appendix D.1, and more visualizations in Appendix D.2.

## D.1. Simple Idea of Using SAM 2

The details about the simple way for adapting SAM 2 for FSS is illustrated in Figure 7, where the image encoder, memory encoder, memory attention and mask decoder constitute to the full SAM 2. The query and support images are forwarded to the image encoder to extract query and support features. Then, support features will be encoded as support memory, based on the support mask prompt. Memory attention consists of 8 consecutive self and cross attentions. During cross attention, the query features will match with support memory, and are dynamically fused with support FG features. Finally, the fused query features are decoded

by the mask decoder to obtain predictions. As explained earlier, the support-query matching here would be incompatible with that of SAM 2, where the former is different-objects matching, and the latter is same-objects matching.

## D.2. More Visualizations

We include more visualizations about pseudo prompts in Appendix D.2.1, and IMR in Appendix D.2.2.

### D.2.1. PSEUDO PROMPTS

We select more testing episodes and depict the input query and support samples, as well as the prior masks (i.e., pseudo mask prompts) and final outputs in Figure 8 and Figure 9, validating: (1) FG prior has more complete FG but much more unexpected BG regions than Disc prior; (2) Although Disc prior masks contain less unexpected query BG regions, so the encoded pseudo Disc memory would have less BG features, they have incomplete FG features, posing challenges to segment complete query FG objects. For example, in the first column of Figure 9, the tole of the ship is wrongly classified as BG objects, because these regions are inactivated in Disc prior; (3) Our method, benefiting from powerful foundation models SAM 2 and DINOv2, can achieve very good results.

### D.2.2. ITERATIVE MEMORY REFINEMENT

More visual impacts of Iterative Memory Refinement (IMR) are depicted in Figure 10 and Figure 11, and we can learn that: (1) IMR can successfully incorporate FG prior and the corresponding memory's FG information into Disc prior and its memory, without propagating too much BG information, as the activated BG regions of the refined Disc prior are consistently less than that of FG prior; (2) IMR cannot help to filter the BG regions initially contained in Disc prior; (3) The worst case of Disc prior after IMR is bounded by FG prior; (4) With the increase of iterations, Disc prior and memory can be fused with more FG information, but also with more unexpected BG information. Therefore, we need to make trade-offs between the incorporated FG and BG information to set the iteration number.

# E. Limitation and Future Direction

There is one failure case/limitation. Kindly remind our method relies on **pseudo mask prompt** to resolve the *incompatible FG-FG matching issue*. In some very difficult examples (all baselines cannot deal with such cases), e.g., the FG and BG are very similar and are quite difficult to distinguish, the generated pseudo mask prompt may be misleading, e.g., the real FG is completely uncovered. This motivates us to further design an error correction module, and we leave it as a future direction.

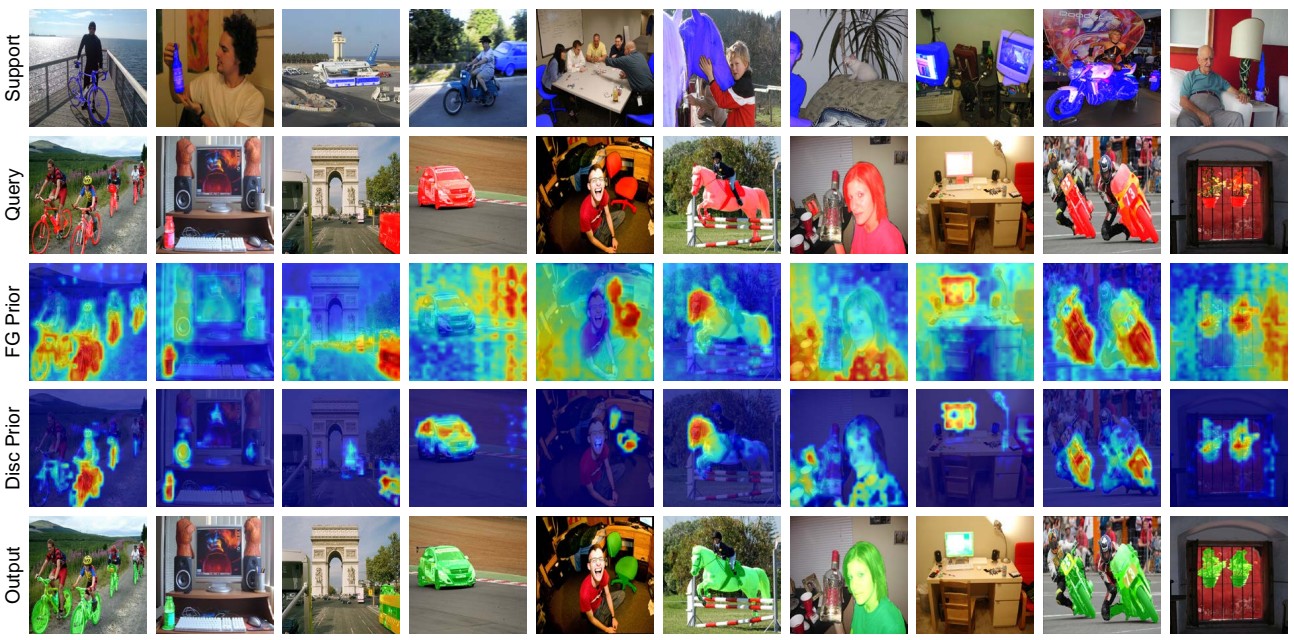

Figure 8. More visualizations of prior masks (i.e., pseudo mask prompts) and outputs.

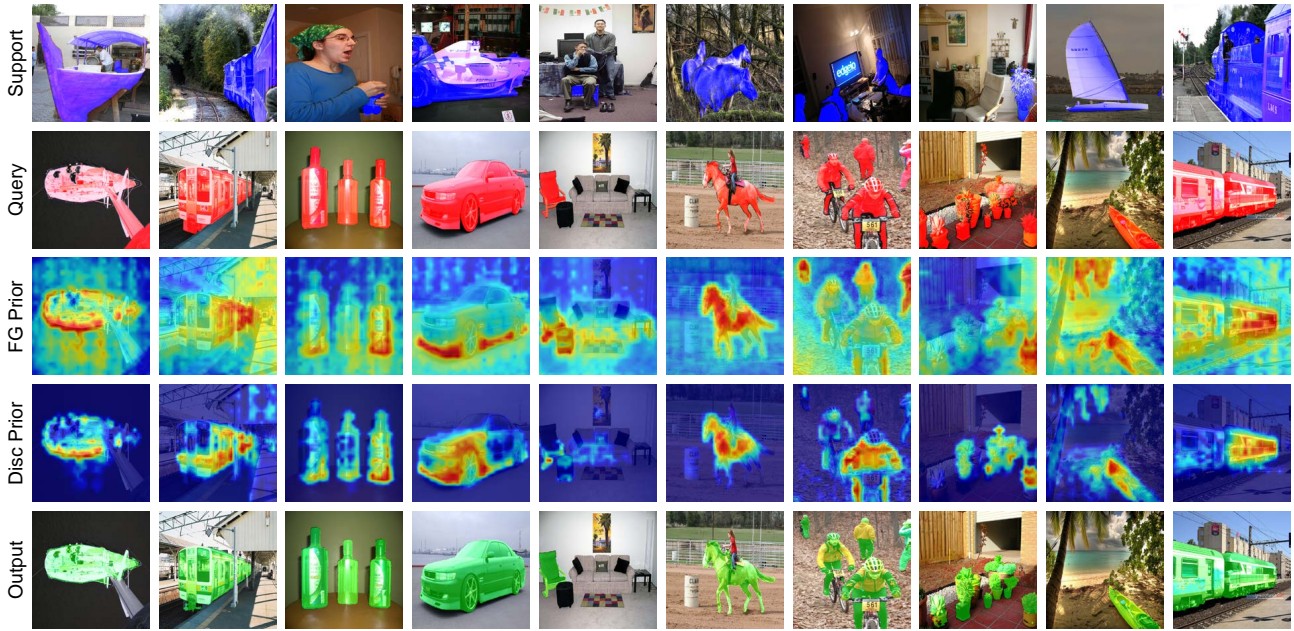

Figure 9. More visualizations of prior masks (i.e., pseudo mask prompts) and outputs.

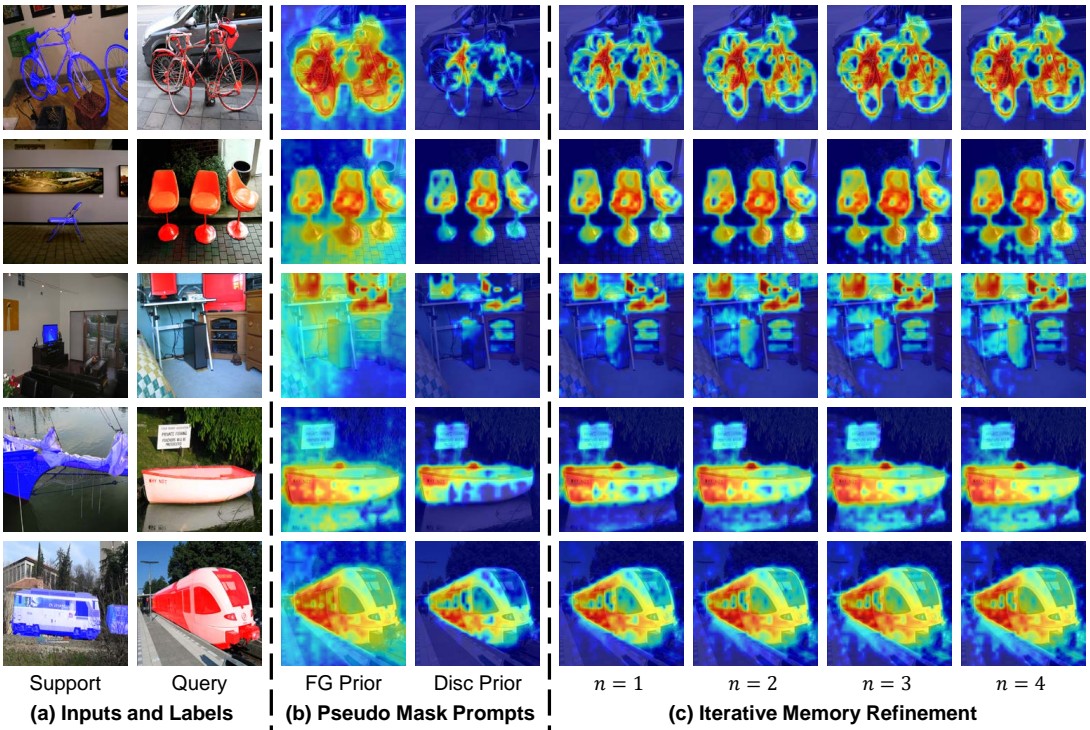

Support   Query       FG Prior   Disc Prior       $n = 1$       $n = 2$       $n = 3$       $n = 4$

**(a) Inputs and Labels**   **(b) Pseudo Mask Prompts**   **(c) Iterative Memory Refinement**

*Figure 10.* More visualizations of Iterative Memory Refinement.

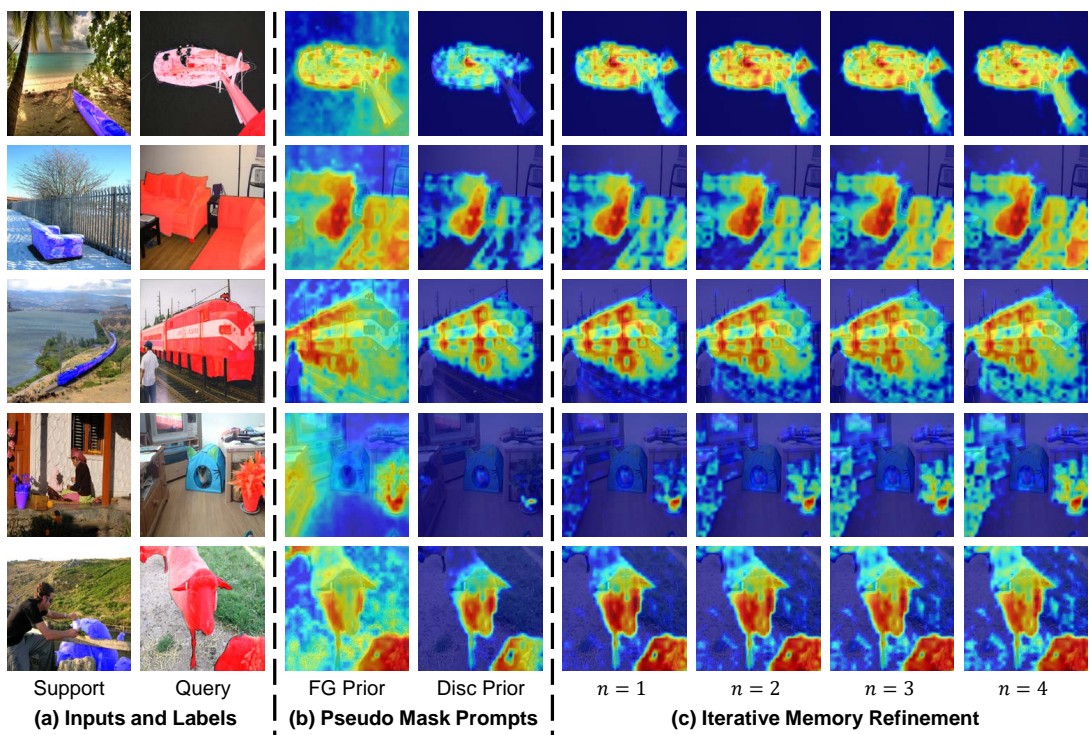

Support   Query       FG Prior   Disc Prior       $n = 1$       $n = 2$       $n = 3$       $n = 4$

**(a) Inputs and Labels**   **(b) Pseudo Mask Prompts**   **(c) Iterative Memory Refinement**

*Figure 11.* More visualizations of Iterative Memory Refinement.

