# OpenReview forum: "Unlocking the Power of SAM 2 for Few-Shot Segmentation"
_ICML.cc/2025/Conference — ICML 2025 poster_

### Official Review · Reviewer_ZVAB · 2025-03-06

**Overall Recommendation:** 4

**Summary:**

This paper utilizes SAM 2 and DINO-v2 to solve the few-shot segmentation problem. The authors first point out that the class-agnostic matching ability of SAM 2 is useful for few-shot segmentation, but SAM 2 focuses too much on the identity of objects, which makes it unsuitable for FSS. To address this issue, the authors propose Pseudo Prompt Generator to generate pseudo query memory and further optimize it using Iterative Memory Refinement and Support-Calibrated Memory Attention. The experimental results show that these methods achieve good performance on PASCAL-5i and COCO-20i datasets.

**Claims And Evidence:**

Yes, the claims made in the submission are supported by clear and convincing evidence. The authors provide a comprehensive analysis of the limitations of SAM 2 in few-shot segmentation and demonstrate that the proposed methods can significantly improve the performance of few-shot segmentation tasks through extensive ablation studies and experiments on PASCAL-5i and COCO-20i datasets.

**Essential References Not Discussed:**

No.

**Experimental Designs Or Analyses:**

Yes, the authors follow traditional FSS settings and conduct extensive experiments to validate the effectiveness of the proposed methods. The ablation studies are also well-designed and comprehensive.

**Methods And Evaluation Criteria:**

the proposed methods make sense for the few-shot segmentation problem. The authors introduce a novel approach that leverages the class-agnostic matching ability of SAM 2 to address the limitations of few-shot segmentation. The evaluation criteria are also well-defined, with experiments conducted on PASCAL-5i and COCO-20i datasets to validate the effectiveness of the proposed methods.

The only limitation is that the proposed methods are not tested on more challenging datasets to evaluate the generalization ability of the models, such as LVIS-92i or cross-domain datasets.

**Other Comments Or Suggestions:**

In line 31, 'but some unexpected query background' seems like 'but' should be 'and'.

**Other Strengths And Weaknesses:**

Strengths:
1. The paper's logic is clear, with the authors first pointing out some limitations of SAM 2 in FSS and then proposing several reasonable modules to address these issues. The design of these modules may have an impact on future FSS research.
2. The experimental results are solid, with FSSAM showing significant improvements over the previous SOTA on PASCAL-5i and COCO-20i.
3. The ablation studies are comprehensive, demonstrating the effectiveness of each module and exploring the impact of some hyperparameters.

Weaknesses:
As I mentioned earlier, my main concern is the generalization ability of the model.
1. Previous works such as Matcher and GF-SAM achieved good results without additional fine-tuning on COCO, while FSSAM requires fine-tuning of some modules, which may affect the model's generalization ability. Is fine-tuning these modules necessary? If not fine-tuned, how much will the model's performance be affected?
2. The authors should test the model on more challenging datasets to validate the generalization ability of the model. For example, LVIS-92i, One-shot Part Segmentation, or some cross-domain datasets. (see Matcher, GF-SAM)

**Questions For Authors:**

Instead of using DINO-v2, VRP-SAM using a trainable resnet also achieves competitive results. Can FSSAM use a trainable resnet instead of DINO-v2? Or, What will happen if FSSAM unfreezes the DINO-v2 backbone and fine-tunes it?

**Relation To Broader Scientific Literature:**

This paper is related to how to leverage foundation models to boost downstream vision tasks.

**Theoretical Claims:**

Not applicable, as the paper does not contain any theoretical claims or proofs.

---

> ### Author Rebuttal · Authors · 2025-03-31
>
> > Evaluation on LVIS-92$^i$.
>
> Thanks for this precious suggestion, conducting evaluation on LVIS-92$^i$ that includes 920 classes can indeed show the **excellent generalizability** of our method. We select both Matcher and SINE (A Simple Image Segmentation Framework via In-Context Examples, NeurIPS'24) for comparisons. Since SINE is trained with COCO (80 classes) and directly test on LVIS-92$^i$, we directly take our trained FSSAM on COCO-20$^0$ (trained with 60 classes, under 1-shot setting) to perform evaluation on LVIS-92$^i$. Following Matcher and SINE, each fold comprises 2,300 testing episodes, and the 1-shot results are shown as follows:
>
> |Method|Matcher|SINE|Ours|
> |-|-|-|-|
> |92$^0$|31.4|28.3|**34.7**|
> |92$^1$|30.9|31.0|**37.8**|
> |92$^2$|33.7|31.9|**37.2**|
> |92$^3$|38.1|34.6|**41.1**|
> |92$^4$|30.5|30.0|**33.9**|
> |92$^5$|32.5|31.9|**38.1**|
> |92$^6$|35.9|32.2|**40.6**|
> |92$^7$|34.2|33.7|**38.9**|
> |92$^8$|33.0|30.6|**36.9**|
> |92$^9$|29.7|27.8|**33.8**|
> |Mean|33.0|31.2|**37.3**|
> |FB-IoU|66.2|63.5|**68.4**|
>
> We can observe our FSSAM consistently outperform Matcher and SINE in all folds, showing **excellent generalizability** (COCO-20$^i$ $\to$ LVIS-92$^i$). We will include these comparisons in a newer version.
>
> > Is fine-tuning SAM 2 necessary?
>
> Yes, fine-tuning is necessary, and the reasons are as follows:
> 1. Original SAM 2's memory attention is trained for **same object matching**, while the matching of FSS is between *different query and support FG objects*, which we call it *incompatible FG-FG matching*. As we can observe from the table, with (w/) or without (w/o) fine-tuning (FT) show prominent performance gap.
> 2. FSSAM can consistently outperform SAM 2 by large margins, since we design modules to resolve the issue. However, FSSAM still requires fine-tuning, because the pseudo mask prompt is *inaccurate*, e.g., it covers *incomplete FG regions* and *unexpected BG regions*, which is different from SAM 2's mask prompt.
>
> |Method|5$^0$|5$^1$|5$^2$|5$^3$|Mean|
> |-|-|-|-|-|-|
> |SAM 2 w/o FT|49.1|43.7|51.1|35.0|44.7|
> |SAM 2 w/ FT|71.8|74.4|71.6|59.9|69.4|
> |FSSAM w/o FT|61.9|59.8|61.0|51.4|58.5|
> |FSSAM w/ FT|81.6|74.9|81.6|76.0|81.0|
>
> Besides, in our response to your previous question, our FSSAM shows **excellent generalizability**, since our performance on LVIS-92$^i$ (the model is trained on COCO-20$^0$ and directly tested on LVIS-92$^i$) consistently outperform other methods by large margins.
>
> > Typo in Line 31.
>
> Thanks for your careful checking! We have corrected this typo.
>
> > Can FSSAM use a trainable resnet instead of DINO-v2? What will happen if FSSAM unfreezes the DINO-v2 backbone and fine-tunes it?
>
> Kindly remind DINOv2 is only responsible for generating pseudo mask prompt without learning, so it can be replaced as pretrained ResNet50, and there is no need to make it trainable. When we replace DINOv2 as ResNet50, the 1-shot mIoU on PASCAL-5$^i$ is 74.1, which is still much better than other methods (in Table 1). kindly note that FounFSS also uses DINOv2, and our performance is 4.2\% better.
>
> For the second question, we can fine-tune DINOv2, but we need to design an extra learning objective to encourage better pseudo mask prompt, which may raise *inductive bias* issue (i.e., *degrading the generalizability*), since the setting of FSS is **training on some base classes while testing on unseen classes**. Besides, the *training cost would be much heavier*, since DINOv2-B has 86M parameters.

---

> > ### Comment · Reviewer_ZVAB · 2025-04-03
> >
> > Thanks for the detailed response and the additional experiments on LVIS-92, which effectively address my concerns regarding generalization. I also appreciate the clarification on the necessity of fine-tuning and the role of DINOv2. I will increase my score to 4.
> >
> > In addition, after reviewing Reviewer HtbM’s comments, I would like to share that I am also very familiar with related works such as SegGPT, Matcher, and SINE. I agree with the authors that the comparison with SINE may not be entirely fair, as SINE benefits from in-domain training data. The strong performance of FSSAM on LVIS-92i provides compelling evidence of its superior generalization ability. That said, including results on one-shot part segmentation would further strengthen the paper.

---

> > > ### Author Response · Authors · 2025-04-04
> > >
> > > Sincerely thanks for your recognition of our work, as well as your explanation to Reviewer HtbM's comment! Following your suggestions, we further conduct evaluation on on-shot part segmentation dataset PASCAL-Part. Specifically, we still use the model trained on COCO-20$^0$ to directly perform evaluation, and the 1-shot results are as follows.
> > >
> > > |Method|F-0|F-1|F-2|F-3|mIoU|
> > > |-|-|-|-|-|-|
> > > |Matcher|**37.1**|56.3|32.4|45.7|42.9|
> > > |Ours|29.5|**73.1**|**34.9**|**48.0**|**46.4**|
> > >
> > > We can observe that our method still outperforms better than baseline Matcher in all folds except fold 0. To figure out the reasons, we visualize the test samples of fold 0, and find that DINOv2 cannot generate very good pseudo mask prompts for these classes, leading to relative low scores. Particularly, in fold 1, our method can surpass Matcher by 16.8\%, and we attribute it to the fact that DINOv2 can generate quite good mask prompts for the classes in this fold. We will include such evaluation in our paper.

---

### Official Review · Reviewer_HtbM · 2025-03-10

**Overall Recommendation:** 3

**Summary:**

This paper leverages SAM2’s strong matching ability to do the few-shot segmentation. Considering the matching of SMA2 is for sam2-object matching, the paper introduces Pseudo Prompt Generator (PPG) to generate pseudo query memories, and further design Iterative Memory Refinement (IMR) to supplement this memory with more query FG features, and devise a SupportCalibrated Memory Attention (SCMA) to mitigate the side-effects of unexpected BG features in pseudo memory.

**Claims And Evidence:**

The claims are clear and convincing.

**Essential References Not Discussed:**

The paper didn’t discuss a very related paper: [NeurIPS'24] A Simple Image Segmentation Framework via In-Context Examples. And the evaluation result couldn’t beat the results in SINE [2]. (see Table 1 in SINE)

[2] Liu, Yang, et al. "A Simple Image Segmentation Framework via In-Context Examples." The Thirty-eighth Annual Conference on Neural Information Processing Systems.

**Experimental Designs Or Analyses:**

The experimental designs and analyses (for the ablation study) look good to me. Please see the weakness for other issues

**Methods And Evaluation Criteria:**

The paper presents results on COCO and PASCAL, which are standard FSS datasets. However, it would be valuable to also evaluate on LVIS-92$^i$[1], which is a more challenging benchmark for evaluating the generalization of a model across datasets based on LVIS.

[1] Liu, Yang, et al. "Matcher: Segment anything with one shot using all-purpose feature matching." arXiv preprint arXiv:2305.13310 (2023).

**Other Comments Or Suggestions:**

NA

**Other Strengths And Weaknesses:**

**Strengths**
- Effectively leverages SAM 2’s memory bank matching for few-shot segmentation.
- Comprehensive ablation studies validate each proposed module.

**Weaknesses**
- The pipeline is relatively complex, and the performance seems to fall behind SINE, which is simpler and can perform more tasks. (this is my major concern).
- The paper lacks a discussion of failure cases and potential limitations.
- The visualization results are insufficient. While most of them are effective—such as the comparisons between FG Prior and Disc Prior, as well as the memory refinement across different iterations—the model's results are not compared with segmentation results from other prior models (as well as the original sam2 model's result). Please include these additional visualizations.

**Questions For Authors:**

- As the paper points out, the number of memory iterations should be traded off for better performance. Is there a way to find the optimized iteration for each case adaptively?
- This work primarily focuses on refining the mask memory. Given the strong prompting capability of SAM2, would it be possible to generate prompt points on the target object region and use these prompts in the mask decoder to obtain the final output?

**Relation To Broader Scientific Literature:**

The paper explores the potential of SAM2 for few-shot segmentation, leveraging its strong matching capability and extending it beyond object-specific matching to include class-level matching across different identities.

**Theoretical Claims:**

The equations look good to me.

---

> ### Author Rebuttal · Authors · 2025-03-31
>
> > Evaluation on LVIS-92$^i$.
>
> Conducting evaluation on LVIS-92$^i$ (920 classes) can show the **excellent generalizability** of FSSAM. We select both Matcher and SINE for comparisons. Since SINE is trained with COCO (80 classes) and directly test on LVIS-92$^i$, we directly take our trained 1-shot FSSAM on COCO-20$^0$ (60 classes) for fair evaluation. Following baselines, each fold comprises 2,300 testing episodes, and the 1-shot results are as follows:
>
> |Method|Matcher|SINE|Ours|
> |-|-|-|-|
> |92$^0$|31.4|28.3|**34.7**|
> |92$^1$|30.9|31.0|**37.8**|
> |92$^2$|33.7|31.9|**37.2**|
> |92$^3$|38.1|34.6|**41.1**|
> |92$^4$|30.5|30.0|**33.9**|
> |92$^5$|32.5|31.9|**38.1**|
> |92$^6$|35.9|32.2|**40.6**|
> |92$^7$|34.2|33.7|**38.9**|
> |92$^8$|33.0|30.6|**36.9**|
> |92$^9$|29.7|27.8|**33.8**|
> |Mean|33.0|31.2|**37.3**|
> |FB-IoU|66.2|63.5|**68.4**|
>
> We can observe our FSSAM consistently outperform Matcher and SINE in all folds, showing **excellent generalizability** (COCO-20$^i$ $\to$ LVIS-92$^i$). We will include these results in a newer version.
>
> > (Major concern) Compare with SINE [2] on PASCAL-5$^i$ and COCO-20$^i$.
>
> SINE CANNOT be fairly compared on these datasets:
> 1. FSS models are trained on some **base classes**, then tested on **unseen classes**, i.e., the testing samples are **out-domain** ones.
> 2. In Section 4.2 of SINE, authors mention "SINE is trained with all data of COCO". Specifically, COCO and PASCAL have 80 and 20 classes, and COCO's classes include PASCAL's classes. Since SINE is trained on whole COCO, **it has learned to deal with all classes of PASCAL and COCO during training**, i.e., the testing samples are **in-domain** ones. Since SINE has been trained with the test classes (in each fold of FSS), its scores will naturally be higher than FSS methods. Unless we remove COCO from SINE's training set, SINE **can never be fairly compared on PASCAL-5$^i$ and COCO-20$^i$**.
> 3. That being said, fair comparisons can be conducted on LVIS-92$^i$, and the results are included in previous question, where our method surpasses SINE by **6.1\% mIoU**, under the consistent **out-domain** setting of "training on COCO and directly testing on LVIS-92$^i$".
>
> > Failure cases and limitations.
>
> There is one failure case/limitation. Kindly remind our method relies on **pseudo mask prompt** to resolve the *incompatible FG-FG matching issue*. In some very difficult examples (all baselines cannot deal with such cases), e.g., the FG and BG are very similar and are quite difficult to distinguish, the generated pseudo mask prompt may be *misleading*, e.g., *the real FG is completely uncovered*. This motivates us to further design an error correction module, and we leave it as a future direction. We will include some failure cases in a newer version.
>
> > Insufficient visualization.
>
> Thanks for this suggestion! We will include some baselines' results for comparisons in a newer version.
>
> > Can iteration number (in IMR) be adaptive?
>
> The iteration number can theoretically be adaptive, yet there exist some challenges, and we uniformly fix it as 3 (see Table 4). Let's recall (1) the impact of IMR, and (2) when to use more iterations.
>
> For (1), IMR will **make the incomplete FG regions in pseudo mask prompt complete**, but at risk of *introducing unexpected BGs (noises)*. Generally, using more iterations can make FG regions more complete, but introduce more noises, so trade-offs should be made to determine the iteration number.
>
> For (2), we dive into the test samples, and find that **difficult samples, containing either multiple FG objects or complex BG, require larger iteration number**. The initial pseudo mask prompt of these difficult samples can **only cover limited FG regions**, so more iterations are required to make FG regions complete (to be a better prompt).
>
> Hence, whether the iteration number can be adaptive depends on if such difficult samples can be automatically identified or not. Unfortunately, determining the difficulty of each case is not trivial, i.e., we cannot either *measure the number of FG objects in query images* or *judge whether query FG and BG are easy to distinguish or not*, and we leave it as a future direction.
>
> > Can we use point prompt?
>
> Yes, but it's not a good choice compared to pseudo mask prompt, since:
> 1. Although we can find some points (in query) with largest similarities to support FG features as point prompts, the overall framework is not easy to optimize, i.e., it's hard to determine if a point is the best candidate or not.
> 2. 1 point prompt can only correspond to 1 entity. When there are multiple FG objects in a query image, e.g., 20 people, it's very difficult to automatically find 1 point for each people. If any people is not assigned with point prompt, he will be classified as BG.
> 3. Instead, pseudo mask prompt can be regarded as a special case of point prompt, which can (1) make optimization easier, and (2) include sufficient points (to cover each FG object).

---

> > ### Comment · Reviewer_HtbM · 2025-04-03
> >
> > Thank you to the authors for the detailed response. The explanation of SINE, along with Reviewer ZVAB’s comment, and the results on LVIS-92$^i$ have addressed my major concerns. I also appreciate the thorough responses to the other questions. I will increase my score to 3.

---

> > > ### Author Response · Authors · 2025-04-04
> > >
> > > Sincerely thanks for taking your time to review our paper and provide the valuable suggestions! We will follow your suggestions to include more visualizations, failure cases, and more evaluations on LVIS-92$^i$ in a newer version.

---

### Official Review · Reviewer_pnf2 · 2025-03-12

**Overall Recommendation:** 3

**Summary:**

This paper presents FSSAM, which leverages SAM2 for few-shot segmentation.
FSSAM designs a Pseudo Prompt Generator to generate pseudo query memories, an Iterative Memory Refinement to iteratively refine pseudo query memories, and a Support-Calibrated Memory Attention to suppress background noise.
Extensive experiments demonstrate state-of-the-art performance.

**Claims And Evidence:**

N/A

**Essential References Not Discussed:**

N/A

**Experimental Designs Or Analyses:**

N/A

**Methods And Evaluation Criteria:**

N/A

**Other Comments Or Suggestions:**

N/A

**Other Strengths And Weaknesses:**

Strengths
- The idea of employing SAM2's Foreground-Foreground matching ability in video segmentation to measure feature similarities between query and support in few-shot segmentation is interesting.
- FSSAM designs several effective modules to make SAM2 adapt to few-shot segmentation. Extensive quantitive and qualitative analysis enhances its interpretability.
- The proposed method outperforms the existing SOTA on two benchmarks.

Weaknesses
- FSSAM uses SAM2 and DINOv2, which may have a larger number of parameters than other methods and result in additional computational cost.
- It's an interesting work but lacks substantive discussion. What was the most important finding in this work? Is the main finding that SAM2 can be used for FSS, or can a memory-based video segmentation model be used for FSS through the proposed module?
- The previous approaches usually obtain better results using a larger model. Table 7 in the appendix explores the impact of different backbones on performance. It can be found that using a larger backbone does not bring consistent improvement. Does it indicate that the method has poor scalability?

**Questions For Authors:**

N/A

**Relation To Broader Scientific Literature:**

N/A

**Theoretical Claims:**

N/A

---

> ### Author Rebuttal · Authors · 2025-03-31
>
> > Larger number of parameters than other methods and result in additional computational cost.
>
> Thanks for this comment. Our parameter number is actually much smaller than most of foundation-based FSS methods. We select some methods and summarize their parameter number, learnable parameter number, as well as the 1-shot mIoU scores on PASCAL-5$^i$ as follows:
>
> |Method|#Params (M)|#Learnable Params (M)|mIoU|
> |-|-|-|-|
> |HDMNet|51|4|69.4|
> |AMNet|54|7|70.1|
> |HMNet|62|15|70.4|
> |Matcher|941|0|68.1|
> |VRP-SAM|666|2|71.9|
> |GF-SAM|941|0|72.1|
> |FounFSS|87|1|76.8|
> |Ours|132|11|81.0|
>
> The first 3 rows correspond to classical FSS methods that use ResNet50 as the pretrained backbone, and the remaining methods refer to foundation-based FSS methods that use DINOv2 and/or SAM. For our finalized model, we use DINOv2-B (86M) and SAM 2-S (46M) without extra parameters (kindly remind our proposed modules are parameter-free), and fine-tune part of SAM 2's parameters.
>
> It can be observed from the table:
> 1. Among foundation-based FSS methods, our parameter number is much smaller than most of them, while our performance is consistently much better.
> 2. Compared to classical FSS methods, though we use more parameters, the difference is not as large as expected, while the performance gap is quite prominent, so we believe the additional cost is worthy.
>
> For **computational complexity**, the designed modules will introduce additional **linear complexity** to the original foundation model, which have already been described in "Memory Complexity" of Section 4.2 and 4.3.
>
> Therefore, the computational burden of our FSSAM is reasonable and acceptable.
>
> > Substantive discussion about the most important finding. Is the main finding that SAM 2 can be used for FSS, or can a memory-based video segmentation model be used for FSS through the proposed module?
>
> Our main finding is **memory-based video segmentation model can be used for FSS through the proposed modules**, and we focus on one of the most representative and powerful models, i.e., SAM 2.
>
> In Table 1, baselines Matcher, VRP-SAM and FG-SAM uniformly deploy SAM-L for FSS, where SAM-L (641M) is much larger than the one we deployed, i.e., SAM 2-S (46M).
>
> As shown in the first row of component-wise ablation study in Table 3, naive SAM 2-S cannot outperform any of these baselines, and we attribute it to the facts that (1) our SAM 2-S (46M) is much smaller than their SAM-L (641M), and (2) there exist an *incompatible matching issue* (as introduced in Section 1).
>
> After using our designed modules to resolve this issue, our FSSAM can use much fewer parameters to outperform these foundation-based FSS methods by large margins, showing the effectiveness of our design, serving as our main finding.
>
> > Larger backbone cannot guarantee consistent improvement.
>
> We would like to make the following notes to Table 7:
> 1. We study 3 versions of SAM 2, including S (46M), B (80.8M) and L (224.4M). According to the official Github of SAM 2, *SAM 2-B (80.8M) originally CANNOT outperform SAM 2-S (46M) in 2 out of 3 datasets*, so it's reasonable that adapting SAM 2-B (80.8M) for FSS cannot show comparable performance as SAM 2-S (46M). When we remove SAM 2-B (80.8M), the improvement is stable, e.g., the mIoU is increased from 81.0 (SAM 2-S, DINOv2-B) to 81.5 (SAM 2-L, DINOv2-B).
> 2. DINOv2 is only responsible for generating cosine similarity-based pseudo mask prompt, **whose features will not be directly used in other modules**, thus upgrading DINOv2 from from B (86M) to L (300M) cannot guarantee better performance, i.e., the pseudo mask prompt generated by DINOv2-B is good enough.
>
> In summary, using larger backbone (SAM 2) can guarantee improvement.

---

> > ### Comment · Reviewer_pnf2 · 2025-04-03
> >
> > I thank the authors for responding to my concerns. According to the rebuttal, the authors claim that the main finding is "memory-based video segmentation model can be used for FSS through the proposed modules".
> > I think it should be comprehensively evaluated using different memory-based video segmentation models.
> > So I keep my rating.

---

> > > ### Author Response · Authors · 2025-04-04
> > >
> > > We sincerely thank you for providing precious suggestions, including the initial ones and the latest one of trying different memory-based video segmentation models. We agree with you it would make our proposed modules stronger, as they can be used like "plug-ins" in this way, for memory-based video segmentation models. Unfortunately, we cannot finish training these new models in this short discussion period. That being said, we will incorporate your other comments into our paper first, and include the "plug-in" experiments once complete.

---

### Official Review · Reviewer_n9Lm · 2025-03-15

**Overall Recommendation:** 3

**Summary:**

The paper introduces the Few-Shot Segment Anything Model (FSSAM), a novel method that leverages the powerful matching capabilities of SAM 2 to enhance few-shot segmentation tasks. The authors address the challenge of adapting SAM 2's same-object matching ability to the different-object matching required in few-shot segmentation by proposing the Pseudo Prompt Generator (PPG), which generates pseudo query memories using prior masks to enable compatible matching between support and query features. To further refine these memories and suppress background noise, the paper introduces the Iterative Memory Refinement (IMR) module, which iteratively incorporates more complete foreground features, and the Support-Calibrated Memory Attention (SCMA) module, which suppresses unexpected background features during the attention process. Extensive experiments on PASCAL-5i and COCO-20i benchmarks demonstrate state-of-the-art performance, with significant improvements in the averaged mIoU and FB-IoU compared to previous methods.

**Claims And Evidence:**

The claims made in the submission are generally well-supported by clear and convincing evidence. The authors provide a comprehensive approach to adapting SAM 2 for few-shot segmentation tasks and demonstrate its effectiveness through both quantitative and qualitative results.

**Essential References Not Discussed:**

No

**Experimental Designs Or Analyses:**

The selection of datasets, evaluation metrics, and ablation studies offers a comprehensive assessment of the proposed method's performance.
However, in the "Parameter Study on IMR" analysis, the performance decreases when the iteration n is equal to 4, yet it increases in 5^3. Is there any further analysis?

**Methods And Evaluation Criteria:**

The proposed methods (PPG, IMR, SCMA) and evaluation criteria (mIoU, FB-IoU on PASCAL-5i and COCO-20i) are relevant and appropriate for the problem of few-shot segmentation. They address key challenges in adapting SAM 2 to this task and provide a comprehensive evaluation of the model's performance. The methods are well-supported by both quantitative results and qualitative visualizations, making them suitable for the application at hand.
However, it would be beneficial to include a comparison with versions of these methods that incorporate learnable parameters. This would help demonstrate the impact of learnable parameters on the model's adaptability and performance, and further validate the robustness of the proposed approach in different scenarios.

**Other Comments Or Suggestions:**

There appears to be a discrepancy between the data in Table 4, where the value for n=3 in 5^3 is 76.0, and the corresponding value in Table 1 is 75.9.
Using the same symbol AQQ for different quantities can indeed lead to confusion. To improve clarity, it would be advisable to use distinct symbols in Equation (13).

**Other Strengths And Weaknesses:**

The paper leverages the strengths of SAM 2 to address the challenges of few-shot segmentation. The key contributions include the Pseudo Prompt Generator (PPG) for creating compatible pseudo query memories, the Iterative Memory Refinement (IMR) module for enhancing foreground features, and the Support-Calibrated Memory Attention (SCMA) to suppress background noise during segmentation.
While the experiments are comprehensive and demonstrate performance improvements, there is room for deeper analysis. Specifically, further investigation into the impact of computational trade-offs, and the influence of learnable parameters could provide additional insights and strengthen the overall robustness of the proposed method.

**Questions For Authors:**

No

**Relation To Broader Scientific Literature:**

The paper leverages recent advancements in foundation models, pseudo prompt generation, iterative refinement, and attention mechanisms to address the challenges of adapting SAM 2 for few-shot segmentation. The comprehensive evaluation and ablation studies further validate the effectiveness of the proposed methods.

**Theoretical Claims:**

Two A_QQ appear in Figure 3, and in conjunction with Equation (8), do these instances carry different meanings? Additionally, the correlation between Figure 2, "Overview of FSSAM," and the detailed descriptions of each module in the Methodology section appears somewhat inconsistent. More specifically, the relationship between the illustration of IMR in Figure 3 and the overview presented in Figure 2 seems somewhat confusing.

---

> ### Author Rebuttal · Authors · 2025-03-31
>
> > Computational burden and learnable parameters.
>
> Thanks for your valuable suggestion! We agree it would be better to include the **parameter number** for further comparisons. We select some methods and summarize their parameter number, learnable parameter number, as well as the 1-shot mIoU scores on PASCAL-5$^i$ as follows:
>
> |Method|#Params (M)|#Learnable Params (M)|mIoU|
> |-|-|-|-|
> |HDMNet|51|4|69.4|
> |AMNet|54|7|70.1|
> |HMNet|62|15|70.4|
> |Matcher|941|0|68.1|
> |VRP-SAM|666|2|71.9|
> |GF-SAM|941|0|72.1|
> |FounFSS|87|1|76.8|
> |Ours|132|11|81.0|
>
> The first 3 rows correspond to classical FSS methods that use ResNet50 as the pretrained backbone, and the remaining methods refer to foundation-based FSS methods that use DINOv2 and/or SAM. For our finalized model, we use DINOv2-B (86M) and SAM 2-S (46M) without extra parameters (kindly remind our proposed modules are parameter-free), and fine-tune part of SAM 2's parameters. It can be observed: (1) Among foundation-based FSS methods, our parameter number is much smaller than most of them, while our performance is consistently much better; (2) Compared to classical FSS methods, though we use more parameters, the difference is not as large as expected, while the performance gap is quite prominent, so we believe the additional cost is worthy.
>
> For **computational complexity**, the designed modules will introduce additional **linear complexity** to the original foundation model, which have already been described in "Memory Complexity" of Section 4.2 and 4.3.
>
> Therefore, the computational burden of our FSSAM is reasonable and acceptable.
>
> > Comparison with versions of these methods that incorporate learnable parameters.
>
> This comment locates in "Methods And Evaluation Criteria", sorry but we are confused about "**different versions**", do you mean **using SAM 2 and DINOv2 with different sizes** or something else?
>
> If our understanding is correct:
> 1. FSSAM is built upon SAM 2 and DINOv2 **without additional parameters**, the designed modules are **parameter-free**, so the variants in Table 3 have same (learnable) parameters.
> 2. All parameters of DINOv2 are frozen. For SAM 2, we **fine-tune its memory encoder, memory attention and mask decoder**.
> 3. Different SAM 2 mainly differ in image encoders, and **the learnable parameters of FSSAM are uniformly 11M**, regardless of which size is used.
> 4. We have studied the impacts of different sizes in Table 7. For your convenience, we summarize some statistics as follows.
>
> |SAM 2|DINOv2|#Params (M)|#Learnable Params (M)|mIoU|
> |-|-|-|-|-|
> |S|B|132|11|81.0|
> |S|L|346|11|80.6|
> |B|B|167|11|79.9|
> |B|L|381|11|79.8|
> |L|B|310|11|81.5|
> |L|L|524|11|81.1|
>
> After making trade-offs between computational burden and performance, we use SAM 2-S (46M) and DINOv2-B (86M). For reasons why using larger SAM 2 and DINOv2 cannot guarantee performance gain, please refer to our reponses to **Larger backbone cannot guarantee consistent improvement** of **Reviewer pnf2**.
>
> > Same symbol $A_{QQ}$ in different modules.
>
> Thanks for this comment. They uniformly refer to the mutual similarities between two features, but we do agree it would be much better to differentiate them in different modules, e.g., additionally include module names as superscripts.
>
> > Inconsistency between Figure 2 and detailed description in Section 4.
>
> Thanks for pointing out this issue, we will modify them accordingly.
>
> > Confusing relationship between IMR in Figure 2 and Figure 3.
>
> We guess that the confusion comes from the inconsistent outputs of IMR modules in Figure 2 and 3, we will unify them in a newer version.
>
> > In "Parameter study on IMR", performance decreases when the iteration n is 4, yet it increases in fold 5$^3$.
>
> The test classes of fold 5$^3$ comprise of potted plant, sheep, sofa, train, and tv/monitor, and we find that the samples of this fold are more challenging than those in other 3 folds (e.g., the performance of fold 5$^3$ is consistently the worst among all folds for all methods in Table 1).  Specifically, many samples from this fold include: (1) multiple tiny objects, and (2) complex background (BG), making it hard to distinguish FG and BG. Therefore, the initially generated prior masks in fold 5$^3$ CANNOT cover sufficient FG regions, and **requires larger n** in IMR to **complete FG regions**.
>
> Kindly remind with the increase of n, more unexpected BG regions will also be activated, acting as some noises to prevent from boosting the performance. Therefore, n should not be too large, otherwise, the performance (e.g., that of fold 5$^0$, 5$^1$ and 5$^2$) will decrease.
>
> In Table 4, we conduct experiments to verify the impacts different iteration number, and we uniformly set it as 3 for all folds currently. We think a better way is **adaptively determining the iteration number n for each sample**, and we will leave it as a future direction.
>
> > Discrepancy between the data in Table 1 and 4.
>
> Thanks for your careful checking! We will correct the value in Table 1.

---

> > ### Comment · Reviewer_n9Lm · 2025-04-06
> >
> > The authors have addressed most of my concerns. I will raise my score to 3.

---

> > > ### Author Response · Authors · 2025-04-06
> > >
> > > We appreciate your time spent on reviewing our paper, as well as the valuable suggestions! We are pleased to hear most of your concerns have been addressed, and we will follow your suggestions to include the parameter information and address other issues like typos.

---

### Decision · Program_Chairs · 2025-05-01

**Decision:**

Accept (poster)

**Comment:**

This paper presents FSSAM, which extends SAM 2 and DINO-v2 for few-shot segmentation with a new Pseudo Prompt Generator and two tailored modules. The method shows strong performance on standard benchmarks and demonstrates solid generalization to LVIS-92 and PASCAL-Part. The rebuttal effectively clarified technical points and strengthened the case for the paper’s contributions. The reviewers appreciated the paper's novelty and contribution, and the final ratings are WA, WA, WA, and A. Given its ratings and contributions, I recommend acceptance.